# A bispecific CD40 agonistic antibody allowing for antibody-peptide conjugate formation to enable cancer-specific peptide delivery, resulting in improved T Cell proliferation and anti-tumor immunity in mice

Aman Mebrahtu[1,2,7], Ida Laurén[2,3,7], Rosanne Veerman[2], Gözde Güclüler Akpinar[2], Martin Lord[2,3], Alexandros Kostakis[2,3], Juan Astorga-Wells[4], Leif Dahllund[1,5], Anders Olsson[1,5], Oscar Andersson[1,5], Jonathan Persson[1,5], Helena Persson[1,5], Pierre Dönnes[2,6], Johan Rockberg[1,2,7] ✉ & Sara Mangsbo[2,3,7] ✉

Current antibody-based immunotherapy depends on tumor antigen shedding for proper T cell priming. Here we select a novel human CD40 agonistic drug candidate and generate a bispecific antibody, herein named BiA9*2_HF, that allows for rapid antibody-peptide conjugate formation. The format is designed to facilitate peptide antigen delivery to CD40 expressing cells combined with simultaneous CD40 agonistic activity. In vivo, the selected bispecific antibody BiA9*2_HF loaded with peptide cargos induces improved antigen-specific proliferation of CD8+ (10-15 fold) and CD4+ T cells (2-7 fold) over control in draining lymph nodes. In both virus-induced and neoantigen-based mouse tumor models, BiA9*2_HF demonstrates therapeutic efficacy and elevated safety profile, with complete tumor clearance, as well as measured abscopal impact on tumor growth. The BiA9*2_HF drug candidate can thus be utilized to tailor immunotherapeutics for cancer patients.

Immunotherapy in the form of checkpoint inhibitors has the power to target metastatic cancer, enabling T cell driven cytotoxicity that impacts patient overall survival and can cure severely ill patients. Whilst checkpoint inhibitors harness the power of an existing anti-tumor response, tumor-targeted antibodies, and CAR-T cells depend on antibody-based targeting units and are thus constrained to a limited amount of druggable targets (i.e., surface-expressed extracellular domains of whole proteins) and thereby sensitive to antigen escape.

Cancer vaccines and immune-activating (agonistic) immunotherapies harbor the potential to induce specific anti-tumor immune responses against any tumor antigen of interest, presented by major histocompatibility complex (MHC), which complement checkpoint inhibitors. Novel agonistic antibodies, as well as vaccine formulations have been developed to optimize the therapeutic index[1]. However, lipid-based formulations can trap expanded antigen-specific cells and skew the immune response towards Th2 rather than Th1 responses[2].

[1]KTH Royal Institute of Technology, Department of Protein Science, School of Engineering Sciences in Chemistry, Biotechnology and Health, Stockholm, Sweden. [2]Strike Pharma AB, Uppsala, Sweden. [3]Department of Pharmacy, Science for Life Laboratory, Uppsala University, Uppsala, Sweden. [4]Department of Medical Biochemistry and Biophysics, Karolinska Institute, Stockholm, Sweden. [5]Science for Life Laboratory, Drug Discovery and Development, Stockholm, Sweden. [6]SciCross AB, Skövde, Sweden. [7]These authors contributed equally: Aman Mebrahtu, Ida Laurén, Johan Rockberg, Sara Mangsbo. ✉e-mail: johanr@biotech.kth.se; sara.mangsbo@farmaci.uu.se

Moreover, nanoparticle-based delivery systems suffer from poor bioavailability and pharmacokinetic properties[3,4]. While DNA/mod-RNA-based platforms commonly require incorporation of antigen targeting and adjuvant into the coding sequence to optimize efficacy, unmodified RNA with inherent adjuvant capacities may lead to hampered peptide/protein production in vivo due to its intrinsic immunogenic profile, leading to variability in exposure to the translated peptide/protein chain. Current mRNA-based strategies are additionally limited by the burden of low-cost efficient manufacturing workflows and lack well-established easy to scale-up downstream processes[5].

A strategy to overcome these challenges is to make use of antibody-based delivery systems carrying cargo by conjugation, in an antibody-drug conjugate format (ADC) where the antibody targeting domain itself can impact tumor growth. The application of ADCs has been shown to increase the specificity of antigen delivery and improve T-cell responses[6]. Dendritic cell targets such as DEC-205, Dectin-1, CD207, LOX-, integrin's, and CD40 have been explored wherein the latter demonstrated superior effect at evoking antigen-specific CD8+ T cell response compared to the other investigated targeted surface receptor for directed delivery[6].

Targeting the CD40 receptor with the use of agonistic antibodies mimics the interaction with the natural CD40 ligand (CD40L), promoting Th1 induction by IL-12 secretion of the DCs and is, therefore, an ideal immunostimulant in oncology and can also be utilized to deliver antigen cargo. The CD40-CD40L axis is a key regulator of the adaptive immune response, and agonistic anti-CD40 antibodies have confirmed anti-tumor activity in preclinical models, paving the way for clinical evaluation[7–10]. The agonistic antibody activity is mainly governed by two factors, the binding epitope location on the CD40 protein and the choice of IgG subclass[11].

Infusion of high affinity agonistic anti-CD40 antibodies is limited by a narrow therapeutic window coupled to liver dysfunction, cytokine release syndrome (CRS), and immune exhaustion[10,12,13]. Multiple publications support the dependence on tumor antigen shedding and antigen presentation for optimal efficacy of CD40 agonistic antibodies[6,14]. Combination with chemotherapy or intra-tumoral administration has been evaluated to optimize CD40 stimulation and antigen-presentation[15–20]. To facilitate the dual effect of CD40 targeting and antigen-presentation, several bispecific antibody formats have been explored to couple the agonistic impact to the tumor microenvironment[21–24]. Unfortunately, the limited number of druggable tumor-specific surface proteins makes this approach challenging, ADCs can, however, deliver any neoantigen of interest to the DC compartment for antigen-presentation.

An alternative to traditional ADC-based delivery is to allow for affinity interactions between antibodies and cargo for drug loading (in vitro or in vivo e.g., for theranostics). We earlier identified an alternative coupling strategy to the classical biotin-avidin interaction, which is mainly used as a tool in drug development due to the very immunogenic nature of avidin itself. The identified Adaptive Drug Affinity Conjugate (ADAC) technology was based on a bispecific design carrying an scFv that could bind a synthetic peptide with high affinity, to enable drug cargo delivery to CD40 expressing cells. The goal was to provide a high-affinity interaction with the peptide cargo, thereby allowing for a flexible and rapid cargo loading based on clinical sequencing data and an in-hospital mixing step of patient-specific peptides, as a means to provide a dual vaccination and CD40 activating drug entity in one step. Our earlier data provided proof-of-concept data of such technology using a murine-derived scFv and a known B cell epitope (Fig. 1A). The reported published data showed extended cargo half-life, targeted cargo delivery and improved T cell activation/expansion in vitro and in vivo (Fig. 1A, B)[25]. To reach the clinic the proposed technology requires further work, specifically development

of a non-immunogenic tag, a human(ized) binder along with in vivo efficacy data in tumor models.

Here, we identify a novel CD40 agonistic antibody and reformat it into a bispecific construct with the intention of enabling modular antigen peptide delivery for clinical use. The lead drug candidate targets CD40-expressing cells, and with the simultaneous delivery of peptide cargo, the therapeutic intervention leads to a robust specific anti-tumor immune response in vivo, with an improved safety profile in comparison to the reference anti-CD40 antibody selicrelumab. The proof-of-concept validation of the lead ADAC-based drug candidate (herein named BiA9*2_HF) warrants further preclinical evaluation and future clinical investigation.

## Results

### A novel anti-CD40 monoclonal antibody conferring agonistic activity dependent on epitope and IgG subclass

Phage display was employed for the selection of novel human CD40 (hCD40) binders, based on two human synthetic single chain fragment variable (scFv) libraries[26]. Initial binding screens by ELISA and SPR analysis of selected clones led to 11 variants of interest to reformat into full-length antibodies of the IgG2 isotype. The initial choice of the IgG2 isotype for screening of potential agonistic CD40 engagement was based on our own and other's previously published data, favoring IgG2 over IgG1 for optimal agonistic activity in humans[25,27]. A primary screening for agonistic activity of the 11 anti-CD40 IgG2 monoclonal antibodies (mAbs) was carried out on monocyte-derived dendritic cells (moDCs). The two IgG2 clones, A9 and F4 displayed agonistic activity on par with LPS stimulated cells and a reference anti-CD40 IgG2 (1150) agonistic antibody (with similar affinity binding as our candidates) represented in a heat-map (Fig. 2A) combining the activation markers MHC-II, CD86 and CD83 together with secreted IL-12 levels of moDCs. Interestingly, the heat map illustrates three distinct clusters of the antibodies; one highly agonistic cluster (Ab clones: A9 and F4) on par with LPS, one intermediate agonistic cluster (Ab clones: B8, E7, A1) comparable to the reference anti-CD40 with IgG1 backbone (1150) and a non-stimulatory cluster (Ab clones: B3, A6, F7, E8, B1, G2) similar to the vehicle and isotype stimulated cells. Further, the antibody candidates differ in their binding competitiveness with the natural CD40 ligand (CD40L) to the CD40 receptor. Only one of the clones (A9) with a high agonistic profile did not compete with CD40L, whilst three of the clones (F4, B8 and F7) representing different agonistic clusters partially block the ligand and the rest of the clones from the intermediate and non-stimulatory clusters (E7, A1, B3, B1 and G2) show the complete blocking of CD40L engagement with the receptor (Table 1). Moreover, the anti-CD40 antibody clones were assessed for off-target binding towards baculovirus particles, keyhole limpet hemocyanin, E.coli polysaccharides, insulin, and DNA by ELISA. The top candidate clone A9, had low off-target binding and cross-reactivity to the evaluated antigens whilst the second candidate clone F4 exhibited partial off-target binding to the tested antigens (Table 1). Detailed SPR analysis determined the affinity towards human CD40 at 5.1 nM and the ability to bind cynomolgus CD40 at 4.5 nM (Table 1) for the A9 clone. Subsequent epitope mapping by HDX-MS of the A9 clone revealed the binding site to be in the membrane-distal part of the hCD40 receptor shown in the CD40-CD40L crystal structure, highlighting the mapped epitope topographical hot spots. While the B1 clone, with low agonistic activity has its binding site located in the interface within CRD1-CRD2 receptor domains, exhibited a diverging contact point located in closer proximity to the receptor-membrane surface interface (Fig. 2B).

To confirm selection of the optimal isotype for agonistic activity the selected clone A9 was formatted into three IgG isotypes to evaluate the impact on agonistic activity. Aside from the already generated IgG2 isotype (A9*2), a conventional IgG1 (A9*1) isotype was made together with a hybrid version comprising of the IgG2 hinge region grafted into

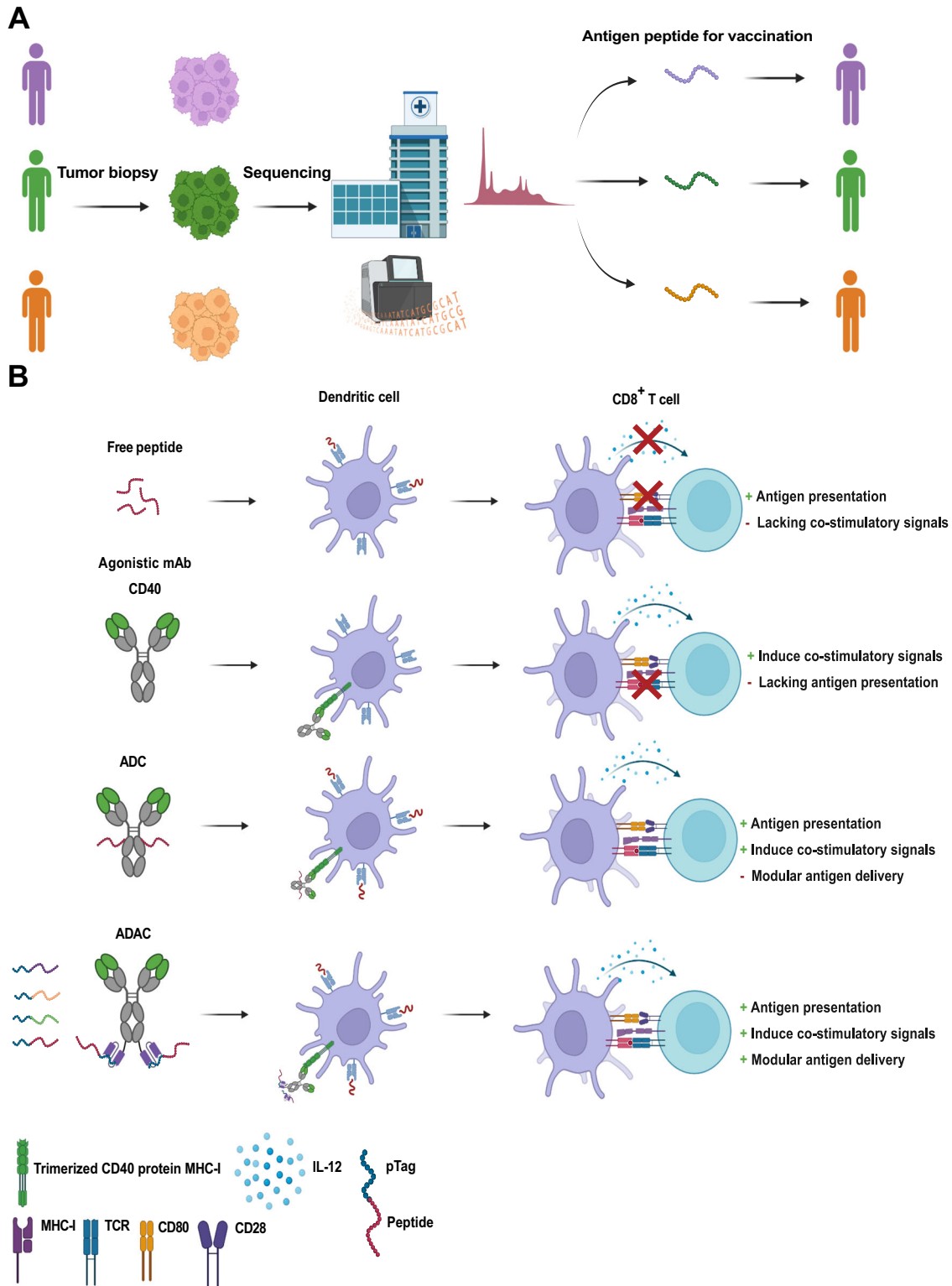

**Fig. 1 | Overview of the principal workflow and advantages of the Adaptable Drug Affinity Conjugate (ADAC) technology. A** Overview of precision immunotherapy using patient-specific identification of tumor neoantigens by sequencing. Patient tumor samples are sequenced for neoantigen determination by in silico prediction. Patient-specific neoantigens are synthesized as peptides for personalized immunotherapy eliciting tumor-specific responses. **B** A modular bispecific antibody, adaptable-drug-affinity-conjugate based antigenic peptide delivery, and subsequent peptide-specific T cell anti-tumor response, compared to other therapeutic immunostimulatory/vaccination modalities and delivery strategies. The bispecific CD40 targeting agonistic antibody drives DC activation, simultaneously allowing for modular delivery of antigenic peptides via a static peptide stretch, the pTag, interacting through a high-affinity interaction with an anti-pTag scFv recombinantly fused to the parental antibody. Illustrations created with Biorender.com.

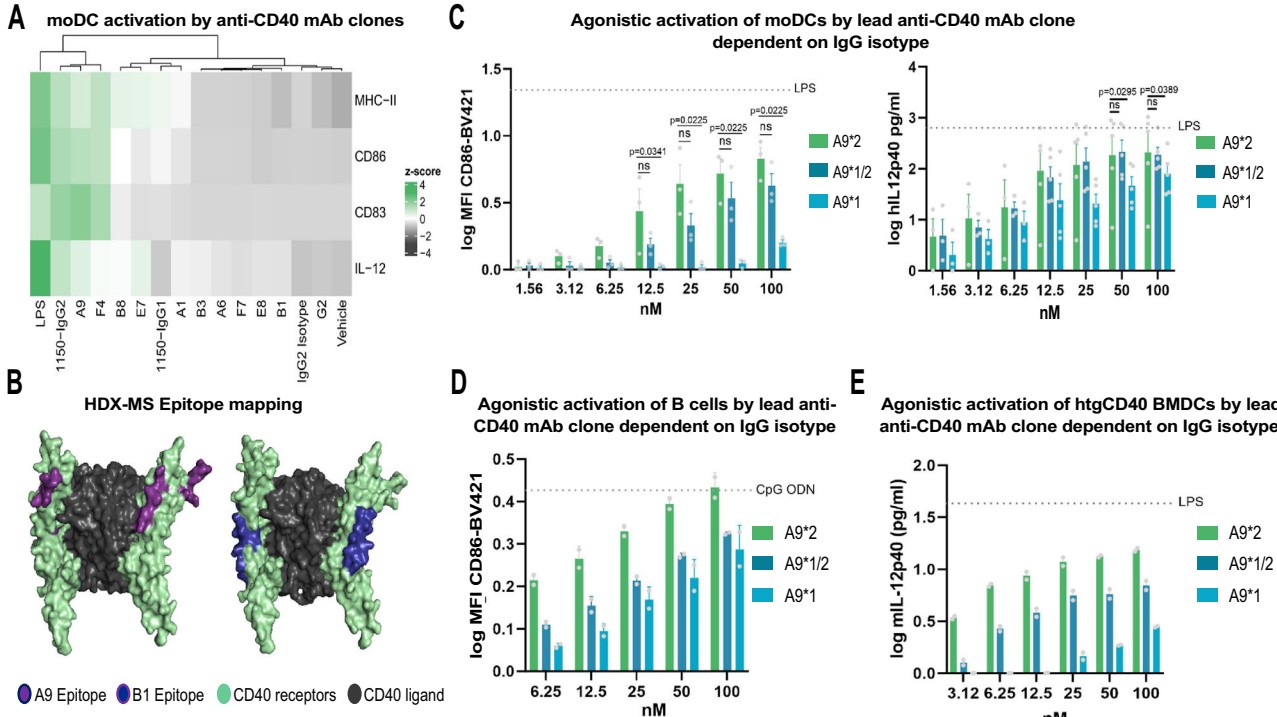

**Fig. 2 | Agonistic activity of novel clones driven by the interplay between epitope, affinity, and IgG subclass. A** The agonistic activity of the anti-CD40 mAbs clones incubated for 48 h with immature moDCs evaluated for upregulation of markers MHC-II, CD86, and CD83 clustered in heatmap together with IL-12 secretion. **B** Epitope mapping of the top agonistic clone A9 and non-agonistic clone B1 determined by HDX-MS illustrated in a CD40L-CD40 crystal structure (PDB 3QD6). **C** Agonistic activity of the mAbs was performed by incubation with immature moDCs for 48 h and evaluated by flow cytometry of CD86 MFI levels (illustrated as background-subtracted log-transformed values) $n = 3$ (independent donors) and IL-12p40 levels $n = 5$ (independent donors pooled from two individual experiments) in the supernatant were quantified by ELISA. *P*-values are shown in the graph (ns = non-significant) and calculated with Kruskal-Wallis with Dunn´s multiple (CD86) and one-way ANOVA with Dunnett´s multiple comparisons test (hIL12p40). **D** Agonistic activity of mAbs evaluated by upregulation of CD86 on isolated CD19[+] B cells via flow cytometry 24 h post-stimulation. $n = 2$ (independent donors, illustrated as the mean of two technical replicates). **E** Immature htgCD40 BMDCs were stimulated for 48 h with the mAbs thereafter, IL-12p40 was quantified by ELISA in the supernatant. $n = 1$ (mean value from two technical replicates). Data is shown as mean ± SEM.

an IgG1 antibody scaffold (A9*1/2) to investigate the impact of the hinge rigidity on the agonistic activity of the chosen lead antibody clone. The IgG2 variant elicited the highest agonistic activity in a concentration-dependent manner of the three generated isotypes by significant upregulation of the activation markers CD86 on moDCs together with secreted IL-12 levels (Fig. 2C). The IgG1 isotype, in line with our previous work and others, displayed a lower level of agonistic activity on moDCs compared to the LPS control and the IgG2 isotype. The hybrid version demonstrated increased agonistic activity although not on par with the IgG2 isotype. A similar pattern was seen by upregulation of CD86 on human B cells (Fig. 2D). Although B cells have high expression of FcγRIIB, enabling cross-linking in trans of the IgG1 antibody, it did not provide agonistic activity compared to the IgG2 subclass. In addition, the observed isotype-dependent agonistic pattern was also established when stimulating human transgenic CD40 bone marrow dendritic cells (htgCD40 BMDCs) (Fig. 2E).

**Generation and humanization of a pTag binding scFv for modular antigen peptide delivery in the ADAC platform**
The anti-pTag scFv constituting the modular peptide-cargo delivery part of the ADAC platform was originally of murine origin and reformatted into a scFv to be employed in the final ADAC bispecific antibody (BiAb) format (Fig. 3A). The murine scFv exhibited nanomolar affinity (4 nM) toward the original 18 amino acids (aa) peptide target sequence but also similar affinity to a c-terminally trimmed sequence down to 12 amino acids (pTag$_{12aa}$) as demonstrated by SPR (Fig. 3B). Given the foreign murine nature of the scFv, a humanization of the scFv was performed by complementarity determining regions (CDR)-

grafting into human germline variable heavy and light (VH/VL) frameworks (Supplementary Fig. 1). To address a putative NG deamidation site located in the HCDR2 region, five additional mutated candidates based on the newly humanized scaffold were generated. A primary evaluation of the expression profile showed an improved protein quality and yield post purification of the humanized scFv compared to the murine version (Fig. 3C), in particular the humanized SG mutated variant. Furthermore, SEC-HPLC and accelerated thermal stability study of the humanized variants post-incubation at 37 °C showed no change in monomeric content compared to untreated samples and the highest Tm for the SG variant at 64.3 °C (Supplementary Table 3 and Supplementary Fig. 2). SPR analysis showed retained binding by the humanized scFv towards the native pTag$_{18aa}$ sequence trimmed down to pTag$_{12aa}$ comparable to the murine version (Fig. 3D), concluding successful reformatting and humanization of the originally isolated anti-pTag antibody clone. Further trimming of the pTag (Supplementary Table 2) was performed down to nine aa with retained binding kinetics and affinity (Fig. 3D), therefore, the pTag$_{9aa}$ was chosen for further studies. The immunogenicity of the trimmed pTag$_{9aa}$ was evaluated after subjecting mice to repeated injections of the BiAb mixed with the pTag$_{9aa}$-E7$_{44-62}$ peptide following lymphocyte isolation and testing T cell reactivity. Stimulation of the isolated lymphocytes with the pTag$_{9aa}$ did not induce T cell responses when evaluated with an IFNγ ELISpot readout, whilst the pTag$_{9aa}$-E7$_{44-62}$ peptide did indeed induce T cell IFNγ production (Fig. 4A). Moreover, ELISpot analysis of long-term co-cultured human moDC and T cells stimulated with pTag$_{9aa}$ alone demonstrated no induction of T cell activity post pTag$_{9aa}$ restimulation (Fig. 4B). In addition, the pTag$_{9aa}$´s potential

**Table 1 | Summary of the monoclonal antibodies' characteristics**

| Selected anti-CD40 antibody variants affinity and binding characteristics | | | | |
|---|---|---|---|---|
| Clone | Kd Fc-hDC40 (nM) | Kd Fc-CynDC40 (nM) | hCD40L Competition | Off-Target Binding |
| A9 | 5.1 | 4.5 | No | Low |
| F4 | 7.6 | 8.1 | Partially | Partial |
| B8 | 13 | 13 | Partially | Low |
| E7 | 6.3 | 13 | Yes | Low |
| A1 | 17 | ND | Yes | Low |
| B3 | 20 | ND | Yes | Low |
| A6 | NA | NA | ND | High |
| F7 | 5.8 | ND | Partially | Low |
| E8 | ND | NA | NA | High |
| B1 | 20 | ND | Yes | Low |
| G2 | 7.8 | ND | Yes | Partial |

*ND* Not Determined.
*NA* Not Available.
The binding affinity against human CD40 protein and cynomolgus CD40 protein was evaluated by SPR. The ability of CD40L binding blocking estimated by SPR and off target binding was assessed by ELISA.

impact on natural peptide processing and presentation was evaluated in an ELISpot assay using the NLV synthetic long peptide (SLP), derived from the CMV pp65 protein, herein referred as $pp65_{489-510}$. The $pp65_{489-510}$ recall response in terms of IFNγ secretion of stimulated human PBMC was equal between the $pp65_{489-510}$ SLP with and without the pTag (no antibody is included) in all nine tested donors. Notably, the $pTag_{9aa}$ alone did not induce any pTag-specific T cell responses with IFNγ levels lower than a three-fold increase above its own negative control (Fig. 4C). Taken together, the SG anti-pTag scFv variant was chosen for adaptation into the ADAC BiAb format in combination with the low immunogenic trimmed pTag.

## Anti-CD40 antibody retains agonistic activity and favorable developability characteristics when reformatted into the optimal bispecific format for use in the ADAC platform

The number of alternative bispecific designs that can be generated within the scope of the ADAC platform range from the choice of fusion point of the scFv to the antibody scaffold, isotype and linker flexibility. Collectively, these intrinsic characteristics of the bispecific molecule could strongly influence the biological efficacy of the drug and, from a developability perspective, the scalability and biophysical properties of the molecule, the latter constituting a well-explored inherent drawback of bispecific antibodies. Hence, an initial screening process of possible bispecific formats was carried out. In total, a set of 13 constructs was generated based on combinations of three main variables: fusion point to either heavy (H) or light (L) chain of the antibody scaffold, linker flexibility (rigid = R and flexible = F), and choice of IgG isotype comprising IgG1 (*1), IgG2 (*2) and aforementioned IgG1/2 hybrid (*1/2). The IgG1 subclass was included as a benchmark comparison relative to IgG2 and hybrid-based formats.

IgG1s are the most common subtype amongst FDA clinically approved antibodies and additionally can provide Fc receptor-mediated cross-linking which has been reported to potentiate agonistic activity with regards to anti-CD40 antibodies. However, within the scope of ADAC, *trans*-binding is unwanted, given that the mode of action is dependent on the internalization of the ADAC molecule and its cargo into APCs to efficiently present the antigen. In summary, the initial screening exhibited expected heterogeneous production and product quality characteristics among the constructs. The scFv fused to the CH3 domain of the antibody scaffold resulted in a 5-10-fold increase in titer, higher post-purification yields, and superior end-product quality exhibited by BLI and SEC compared to light chain (LC) fusion

variants (Fig. 5A). Fusions to the CH3 in the IgG2 format with flexible glycine-serine based linker (BiA9*2_HF) and rigid alpha-helical based linker (BiA9*2_HR) exhibited a monomeric population > 95% post purification and titers comparable to their IgG1 analogs (BiA9*1_HF and BiA9*1_HR). In addition, we observed that improved yield and protein quality of the humanized scFv sustained also in the bispecific ADAC format. To evaluate the potential positive contribution of the humanized scFv to the overall improvement of yield and quality, one additional BiAb based on the original murine version of the anti-pTag scFv fused to the anti-CD40 A9 clone (BiA9*2_HF_mscFv) was generated and compared head-to-head to the bispecific produced with the humanized version of the scFv. As hypothesized, the construct demonstrated inferior product titer and quality, containing an 85.3% monomeric population post purification (Fig. 5A).

The agonistic activity of the IgG2-based CH3 fusions with the flexible and rigid linker, respectively, on moDCs were evaluated in a similar fashion as previously described, benchmarking against the parental anti-CD40 antibody. Illustrating retained agonistic ability when converted into the bispecific format in terms of upregulation of CD86 (Fig. 5B) and secreted IL-12 levels (Fig. 5C) and additionally confirmed by upregulation of CD86 on B cells (Fig. 5D). To reduce immunogenicity, immunogenic regions were screened for T cell epitopes in silico (Supplementary Fig. 3A). The deimmunized variant contains less than 50% of predicted CD4+ T-cell epitopes compared to the wildtype variant introduced by a single point mutation, without hampering the binding ability to the CD40 protein (WT Kd = 3.4 nM, DI = 1.6 nM, Supplementary Fig. 3B) or the agonistic activity (Supplementary Fig. 3C).

In conclusion, fusion to the CH3 domain of the antibody scaffold and an IgG2 isotype based on earlier screenings served as the most optimal format fulfilling quality control (QC) developability criteria and initial biological activation on par with the parental anti-CD40 antibody. The modularity of the ADAC platform's ability to enable variable antigen peptide delivery without hampering the affinity interaction between the scFv and the pTag was evaluated by SPR. Analysis demonstrated retained binding to the $pTag_{9aa}$ synthesized with four different antigen peptide sequences, $pp65_{489-510}$, $gp100_{20-39}$, $OVA_{323-339}$, $E7_{44-62}$, and cargo-free $pTag_{9aa}$ (Fig. 5E). Kinetic SPR experiments were carried out to confirm affinities of the humanized scFv to the trimmed pTag synthesized with clinically relevant peptides (peptides spanning immunogenic KRAS-derived peptides for human drug candidate use). The selected BiAb format BiA9*2_HF was immobilized onto the surface of a sensor ship, and binding was evaluated with the selected peptides by obtained sensorgrams injecting four concentrations in duplicate (1.23, 3.70, 11.1, and 33.3 nM) over the sensor ship. The affinity toward the $pTag_{9aa}$-$KRAS_{G12V}$ peptide was calculated to $3.0 \times 10^{-11}$ M and for $pTag_{9aa}$-$KRAS_{G12D}$ to $4.7 \times 10^{-10}$ M (values may be underestimated due to very slow off rate). To confirm that no off-target binding is present for the final protein drug candidate BiA9*2_HF, the retrogenix array screening was performed, confirming specificity to CD40 while no other off-target hits were identified (Supplementary Table 4).

## The ADAC technology induce peptide antigen-specific T cell expansion in vivo in a pTag-dependent manner

Next, we assessed the ability of the peptide-delivery capacity of the bispecific design to improve antigen-specific CD8+ and CD4+ T cell expansion in vivo. The bispecific design exerts dual functionality by DC activation and antigen delivery, thus improving the T cell priming and expansion. Agonistic anti-CD40 antibodies are known to enhance the cross-presentation of exogenous antigens onto MHC-I for induction of CD8+ T cell activation. To induce antigen-specific CD8+ T cell proliferation and evaluate the effect of the length of the pTag, adoptive transfer of PMEL derived immune cells expressing the congenic marker Thy1.1 were used. The lead bispecific format (BiA9*2_HF) together

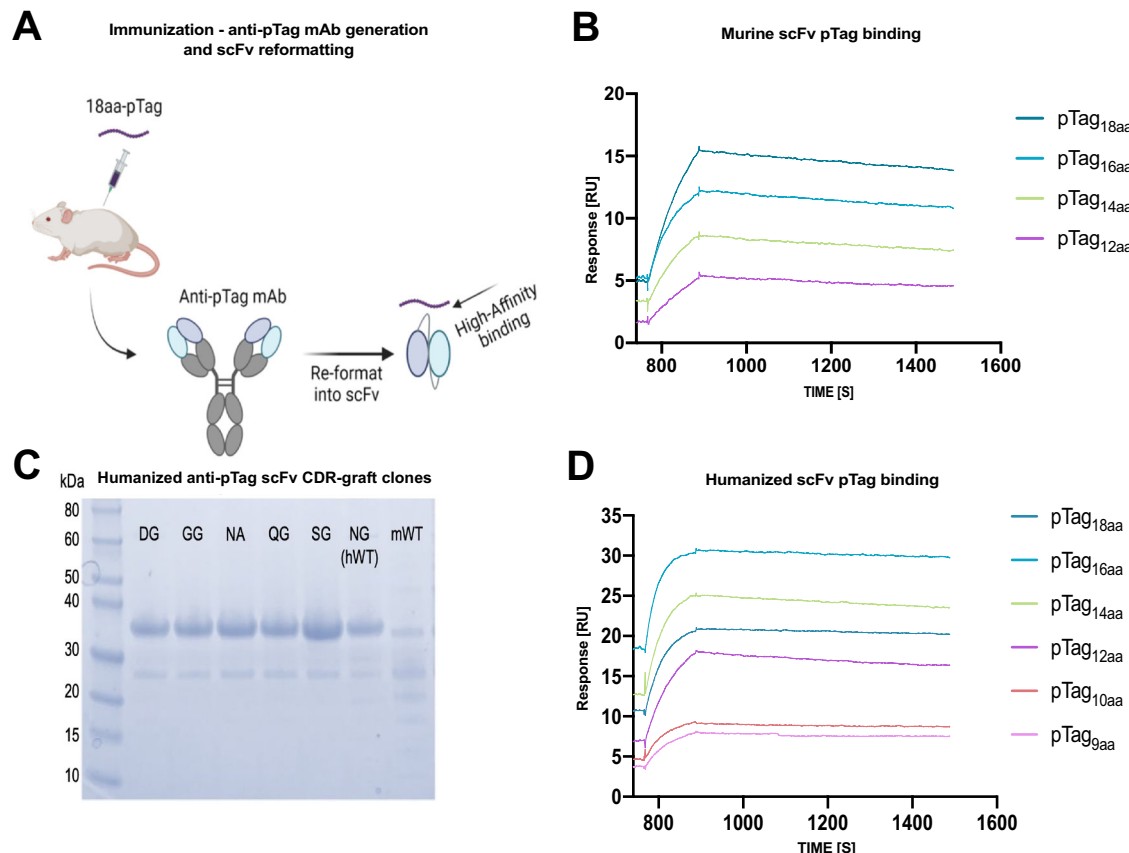

**Fig. 3 | Optimization of anti-pTag scFv and pTag peptide length. A** A 18aa-long unstructured peptide was used to generate a murine-derived scFv. The generated hybridoma from immunization was sequenced and reformatted to a humanized format. **B** SPR sensorgram showing no significant effect on affinity towards the peptide sequence by scFv when trimmed sequence from original 18aa to pTag$_{12aa}$. **C** SDS-PAGE post expression in E. coli and subsequent Ni-NTA purification of humanized anti-pTag scFv clones with deamidation site removed. Molecular reference size markers stated in KDa. The gel was repeated twice with similar results. **D** SPR sensorgram shows retained binding by humanized single chain clone **SG** to pTag peptide trimmed variants ranging from the original pTag$_{18aa}$ sequence down to pTag$_{9aa}$. Illustration created with Biorender.com.

with the pTag-gp100$_{20-39}$ peptide successfully induced CD8$^+$ T cell expansion in the draining lymph nodes (Fig. 6A). The shortest pTag$_{9aa}$ elicited a significantly higher response in a length dependent matter and was the only pTag that generated proliferating cells in the spleen at the assessed dose. The peptides alone were also evaluated in a dose-response in vitro using co-cultured BMDC and PMEL cells without the addition of the BiA9*2_HF. However, whether the difference in T cell proliferation with the trimming of the peptide length is due to improved intracellular processing by peptide trimming or due to peptide content or quality parameters impacting the results, has not been investigated further.

We further wished to explore the impact on CD4$^+$ T cell activation with CD40 stimulation to confirm if the MHC class II derived antigen-presentation pathway was supported by antibody-mediated peptide delivery to antigen-presenting cells. The ability to induce CD4$^+$ T cell activation was assessed with the OT-II model comparing peptide delivery with and without the pTag$_{9aa}$. The affinity interaction between the pTag$_{9aa}$ and BiA9*2_HF is essential for CD4$^+$ T cell proliferation in the draining lymph nodes, demonstrating a 2-7-fold increase compared to the non-linked antigen peptide (Fig. 6B).

To evaluate the efficacy of the ADAC technology, we compared the CD8$^+$ T cell proliferation with another well-known adjuvant, CpG ODN (henceforward called CpG), in a prime-boost setting. As previously demonstrated, the BiA9*2_HF together with the pTag$_{9aa}$-gp100$_{20-39}$ induced a 2-fold increased proliferation compared to without the pTag$_{9aa}$ and a 3-fold increased proliferation compared to peptide alone in both draining and non-draining lymph nodes.

Furthermore, the group vaccinated with CpG and an equal amount of peptide, as with the bispecific antibody, did not induce proliferation in this model. Increasing the peptide dose 12 times in the CpG group rendered the CpG and the BiA9*2_HF to equal levels of T cell proliferation (Fig. 6C).

## ADAC platform induces pTag dependent and peptide-specific anti-tumor response without any notable adverse effects

In earlier studies, we noted that systemic administration of a low dose (30 μg) agonistic anti-CD40 antibody did not achieve the same anti-tumor efficacy as a local administration at the tumor site, illustrating the reliance on antigen shedding of the tumor for efficacy[18]. To investigate the impact of employing ADAC with co-delivery of antigen, using a low-dose administration scheme we made use of the TC-1 tumor model expressing the HPV E7 antigen. To assess the abscopal vaccine-induced anti-tumor effects, the tumor was placed on the right flank and by parental injection of the therapeutics in the left leg via hock injection. Co-delivery of BiA9*2_HF with pTag$_{9aa}$-E7$_{44-62}$ significantly reduced tumor growth and prolonged the survival in the TC-1 model while the single agents or antibody peptide mixed but not linked did not impact tumor growth (Fig. 7A).

The toxicity and efficacy profile of the BiA9*2_HF was stress tested by benchmarking against the clinically evaluated anti-CD40 agonistic antibody selicrelumab in the TC-1 tumor model. BiA9*2_HF co-injected with pTag$_{9aa}$-E7$_{44-62}$ treated groups were administrated on the non-tumor side via hock injection with 30 μg of BiA9*2_HF and 1.5 μg of pTag$_{9aa}$-E7$_{44-62}$ peptide (1:3 molar ratio). Comparatively, 100 μg of

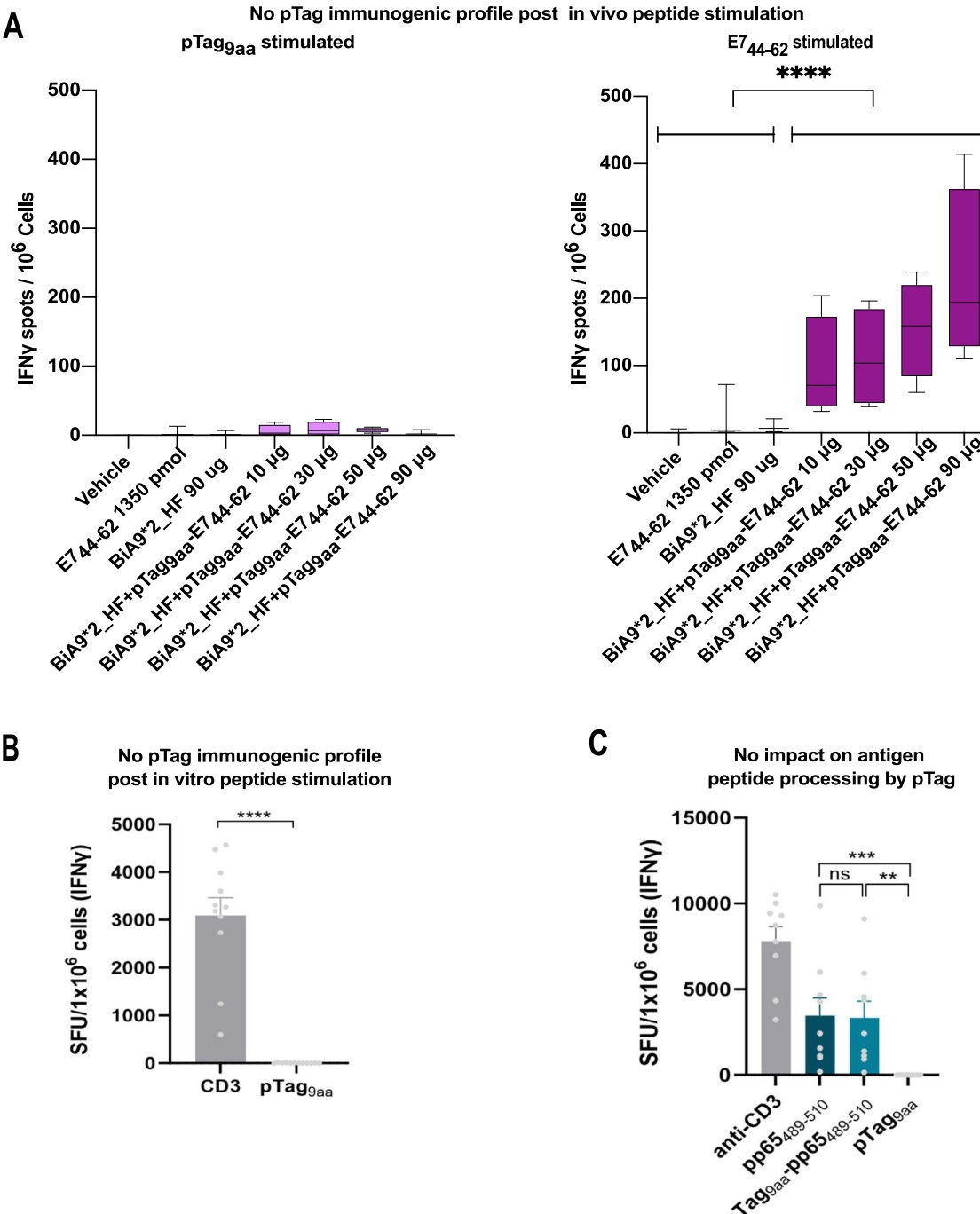

**Fig. 4 | Evaluation of pTag immunogenicity and impact on antigen processing.**
**A** ELISpot analysis of spots of background subtracted IFNγ secretion post-stimulation of isolated lymphocytes from vaccinated mice, with 5 μM of peptides for 22 h. $n = 16$ individual mice with 4 mice in each group. The whiskers show min and max values, and the box extends from 25 to 75 percentiles, with the middle line representing the median. ****$p = <0.0001$ calculated with unpaired Mann-Whitney $t$ test. **B** ELISpot analysis of spots of background subtracted IFNγ secretion post-co-culture of T cells and moDCs treated with 10 μM pTag$_{9aa}$-pp65$_{495-503}$ for 10 days and restimulated with 2 μM pTag$_{9aa}$ or αCD3 for 20 h. CD3 $n = 11$ and pTag$_{9aa}$ $n = 12$ (individual donors pooled from three independent experiments) ****$p = <0.0001$ calculated with unpaired two-tailed $t$ test. **C** ELISpot analysis of spot forming units (SFU) of background subtracted IFNγ secretion post-stimulation with 10 uM of peptides for 24 h $n = 9$ (individual donors pooled from two individual experiments). **$p = 0.0011$, ***$p = 0.0008$ ns = non-significant calculated with Kruskal-Wallis with Dunn´s multiple comparison test. All data are illustrated as mean with ± SEM.

---

selicrelumab was administered by intravenous infusion as clinically used. The overall survival was significantly improved in the BiA9*2_HF co-injected with pTag$_{9aa}$-E7$_{44-62}$ treated group compared to vehicle or selicrelumab treated mice (Fig. 7B). Furthermore, selicrelumab treatment did not induce antigen-specific T cell expansion, at day 17 post tumor inoculation, demonstrated by tetramer staining and analysis by

flow cytometry, whilst local subcutaneous delivery of the BiA9*2_HF linked to the pTag$_{9aa}$-E7$_{44-62}$ peptide significantly did (Fig. 7C).

In terms of toxicity, significant weight loss of the selicrelumab-treated mice was noted following the first two doses, whilst the BiA9*2_HF with pTag$_{9aa}$-E7$_{44-62}$ peptide and vehicle-treated groups remained unchanged (Supplementary Fig. 4A). In addition, the

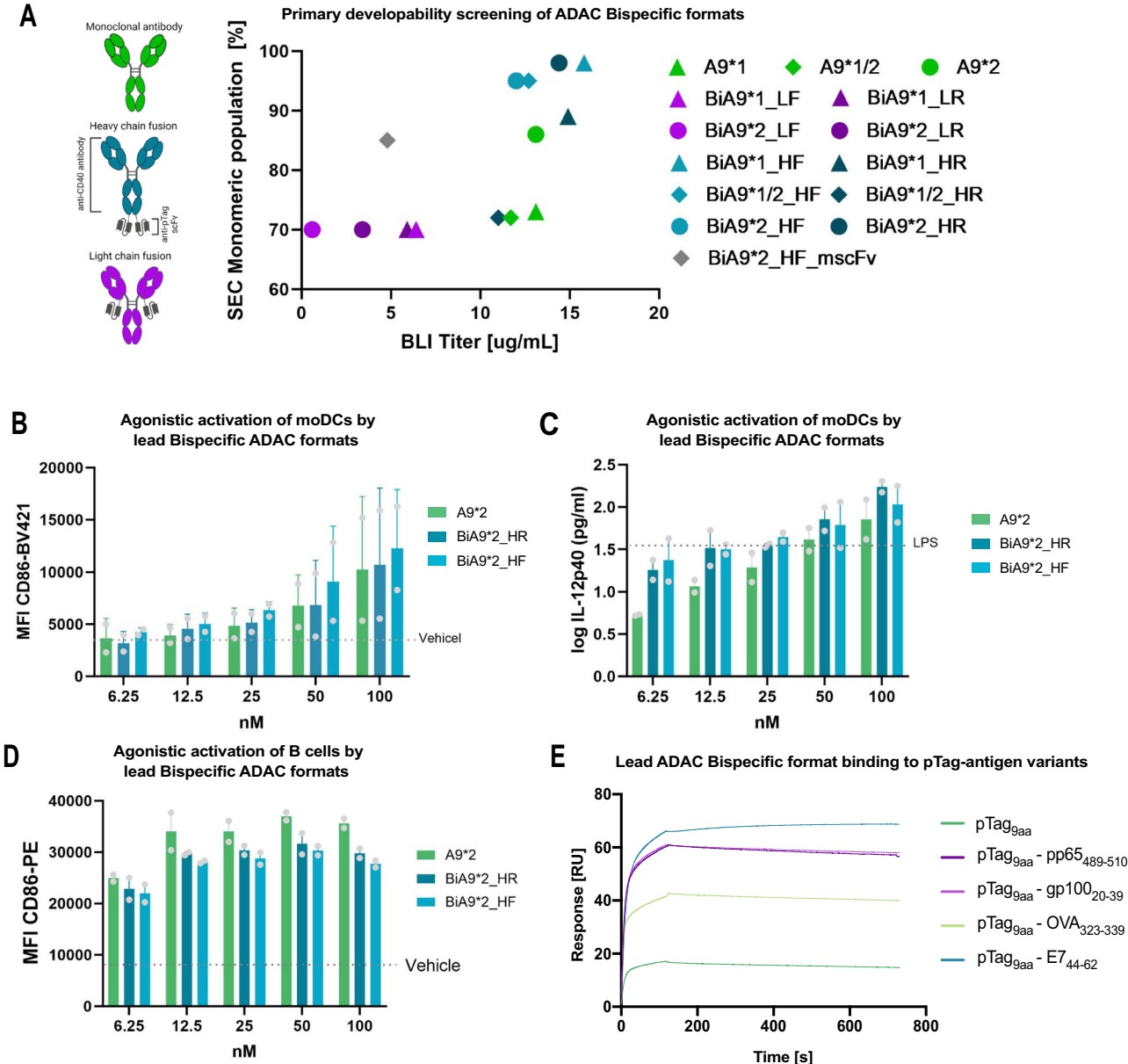

**Fig. 5 | ADAC-adapted Bispecific antibodies based on the novel A9 CD40 agonistic mAb and humanized anti-pTag scFv. A** Developability profile of the generated bispecific constructs adapted for the ADAC platform based on expression titers quantified by BLI and SEC profile post purification. Constructs grouped and colored by fusion point, linker type, and IgG isotype (**B**) MFI of activation marker CD86 evaluated by flow cytometry and (**C**) IL-12p40 levels in supernatant quantified by ELISA on moDCs 48 h post incubation. $n = 2$ (individual donors, illustrated as the mean of two technical replicates). **D** MFI of activation marker CD86 on CD19[+] B cells evaluated by flow cytometry 24 h post incubation with lead BiAb candidates show agonistic activity on par with the parental antibody. $n = 1$ (individual donors, illustrated as two technical replicates). **E** SPR sensogram demonstrating the retained binding and modularity of the ADAC platform when anti-pTag scFv fused to the novel anti-CD40 mAb. Binding to pTag9aa synthesized with four different antigen cargos; pp65489-510, gp10020-39, OVA323-339, E744-62, and non-cargo attached free pTag9aa (**A**) BLI data are based on the mean from triplicate measurements. **B** $n = 2$ (individual donors, illustrated as the mean of two background-subtracted log-transformed technical replicates). **B**–**D** the data are illustrated as mean ± SEM. Illustration created with Biorender.com.

significant increase of AST levels in plasma, indicating liver toxicity, was observed for the selicrelumab treated mice, while low dose s.c and high dose i.v of BiA9*2_HF together with pTag9aa-E744-62 did not elevate AST levels (Fig. 7D). Lowering the antibody dose to 40 μg still led to increased AST levels 24 h post i.v selicrelumab treatment (9908 ng/ml) compared to vehicle (232 ng/ml), BiA9*2_HF i.v (290 ng/ml) and BiA9*2_HF s.c (341 ng/ml). Selicrelumab has been shown to induce platelet depletion in the clinic, thus a human ex vivo ID.Flow assay was used to investigate the effect on platelets as an indication of toxicity. BiA9*2_HF with pTag9aa-KRASG12V did not affect the platelet count (Fig. 7E) nor induce white-blood cell-platelet conjugates compared to

selicrelumab (Supplementary Fig. 4B). Further, BiA9*2_HF cross-react to cynomolgus CD40 with an agonistic effect demonstrated by upregulation of CD86 in a B cell activation assay in vitro (Supplementary Fig. 4C) together with observed systemic immune activation by elevated IL-12/IL23p40 levels in plasma post a second injection in vivo (Supplementary Fig. 4D). In terms of toxicity, no elevation of the liver enzymes AST or ALP was noted (Fig. 7F, G), nor effect on the kidney function (Supplementary Fig. 4E) in NHP after two-three times vaccination. As the human-derived KRAS peptides are not immunogenic in cynomolgus (no epitopes predicted to be presented on cynomolgus MHC molecules), T cell expansion could not be monitored in this

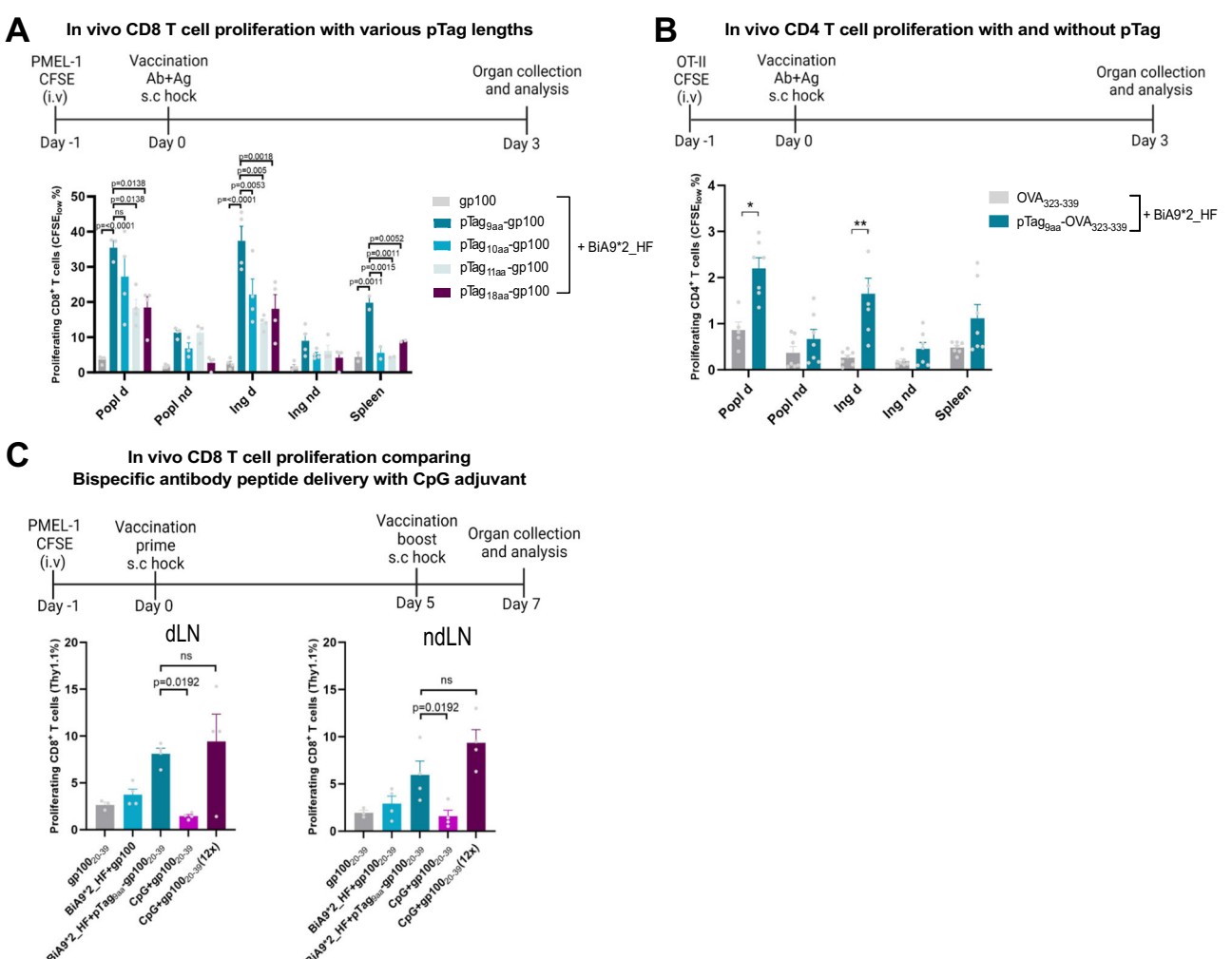

**Fig. 6 | The lead drug candidate BiA9*2_HF induces both CD8⁺ and CD4⁺ peptide antigen-specific T cell expansion in vivo. A** PMEL-1 T cells 72 h post vaccination with BiA9*2_HF (15 pmol = 3 µg) co-injected with a 2.5 molar ratio of the pTag-gp100$_{20-39}$ peptide (37.5 pmol = 0.12 µg) or gp100$_{20-39}$ peptide (37.5 pmol = 0.08 µg) and with length variants of the pTag in the draining popliteal (Popl d), non-draining popliteal (Popl nd), draining inguinal (Ing d), non-draining inguinal (Ing nd) lymph nodes and spleen were quantified by flow cytometry, defined as percentage of CD3⁺, CD8⁺, Thy1.1⁺ and CFSE$_{low}$ (proliferating cells). $n = 20$ (4 mice/group) tgCD40 from Biocytogen. $P$-values are shown in the graph and calculated with one-way ANOVA with Dunnett's multiple comparisons test. **B** OT-II T cell proliferation 72 h post vaccination with the BiA9*2_HF (15 pmol = 3 µg) co-injected with the pTag$_{9aa}$-OVA$_{323-339}$ peptide (37.5 pmol = 0.1 µg) or OVA$_{323-339}$ (37.5

pmol = 0.07 µg) quantified by flow cytometry as percentage CFSE$_{low}$ of the CD3⁺ and CD4⁺ T cells. $n = 14$ (7 mice in each group), tgCD40 from Southampton, data pooled from two independent experiments. *$p = 0.0476$ and **$p = 0.0016$ calculated with unpaired two-tailed $t$ test, (**C**) Proliferating PMEL-1 CD8⁺ T cells post two-time vaccination with the BiA9*2_HF (250 pmol = 50 µg followed by 150 pmol = 30 µg) or CpG (20 µg) combined with the pTag-gp100$_{20-39}$ peptide (375 pmol=1.2 µg) or gp100$_{20-39}$ (375 pmol = 0.83 µg or 4510 pmol = 10 µg) quantified by flow cytometry as percentage of the transferred Thy1.1⁺ cells of the CD3⁺ and CD8⁺ T cells. gp100$_{20-39}$ group $n = 3$, remaining groups $n = 4$, tgCD40 from Biocytogen. $P$-values are shown in the graph (ns = non-significant) and calculated with one-way ANOVA with Dunnett's multiple comparisons test. All data is shown as mean with ± SEM.

model. Altogether these results indicate a low toxicity profile of the BiA9*2_HF in several species.

To evaluate if the ADAC technology could lead to improved anti-tumor effects using a true neoantigen-based antibody-peptide conjugate approach, we employed the MC38 model, together with the identified neoantigen Adpgk[28]. The selected neoantigen Adpgk has not been shown to promote robust anti-tumor responses in earlier published studies[29] using CpG as an adjuvant, interestingly we could confirm that the neoantigen delivered by BiA9*2_HF led to superior anti-tumor control and improved survival compared to a control CpG and peptide treated group (Fig. 8A, B). We also assessed the impact of an elevate dose at the second injection to simultaneously allow for immune microenvironment remodelling and noted a potentiated anti-tumor response with clearing of 8/10 in the BiA9*2_HF with pTag$_{9aa}$-Adpgk$_{295-313}$ treated animals, while the same dose given to animals only receiving the BiA9*2_HF antibody alone or mixed with the neoantigen

carrying a truncated pTag that does not bind the BiA9*2_HF antibody cleared tumors in 4/10 and 1/10 animals respectively (Fig. 8C). Overall survival display superiority of the BiA9*2_HF with pTag$_{9aa}$-Adpgk$_{295-313}$ treated animals compared to the control groups in the same experiment (animals marked with dagger is excluded due to reaching the humane endpoint by either undergoing weight loss greater than 10% or exhibited wounds) (Fig. 8D). To evaluate if it was enough with one single high dose of BiA9*2_HF with pTag$_{9aa}$-Adpgk$_{295-313}$, the vehicle group was administrated a single high dose of the BiA9*2_HF with pTag$_{9aa}$-Adpgk$_{295-313}$ at day 19, this however did not control tumor growth as these mice progressed in tumor growth over time.

Collectively the in vivo data demonstrate that BiA9*2_HF exhibits low toxicity and co-dosing of BiA9*2_HF together with pTag9aa linked antigen induces an antigen-specific T cell expansion in a pTag dependent manner, inducing robust anti-tumor responses in two different tumor models.

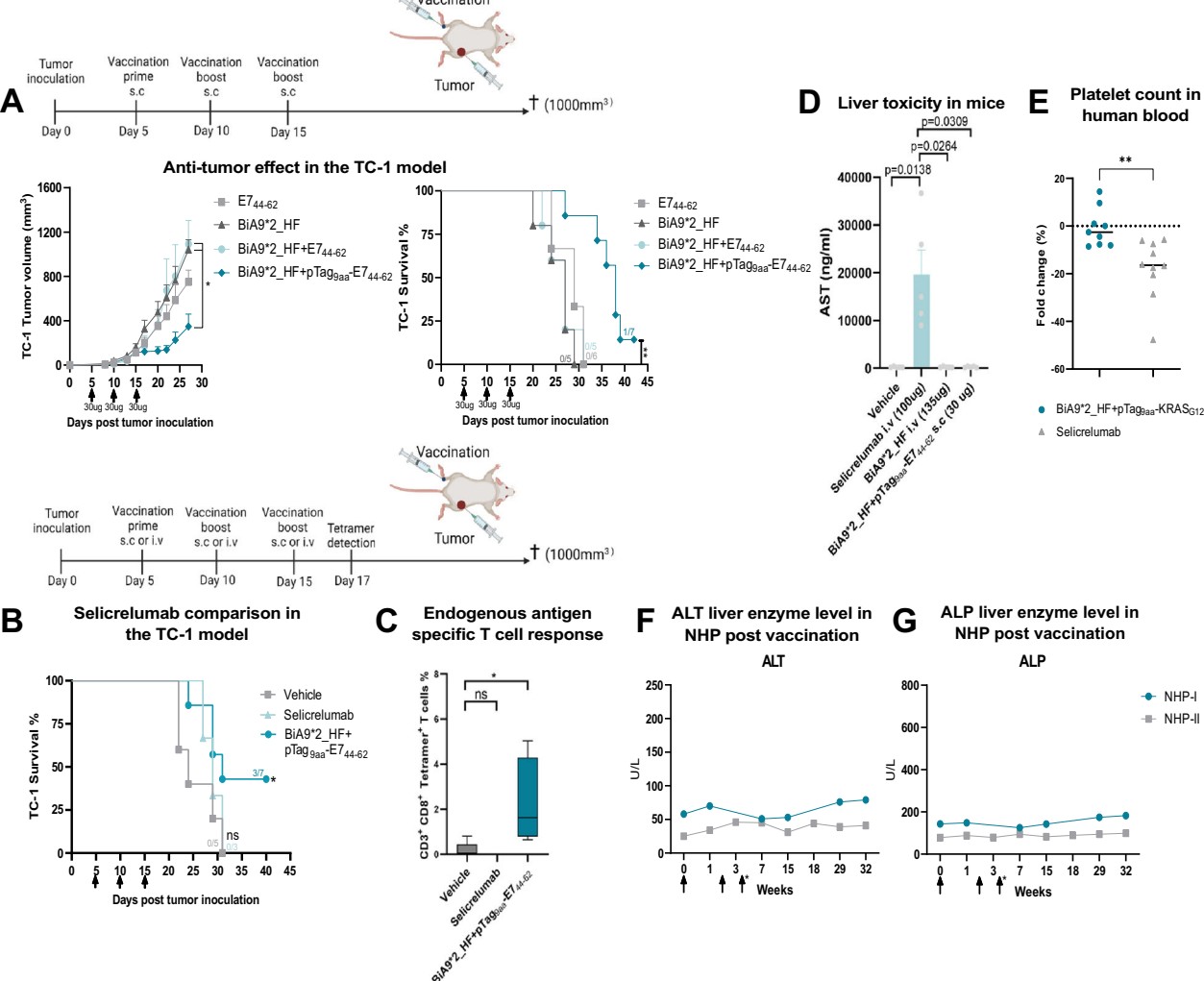

**Fig. 7 | Low dose BiA9\*2_HF loaded with tagged E7 derived tumor antigen induces anti-tumor responses on distant growing TC-1 tumor with minimal toxicity effects. A** Tumor volume over time and survival analysis in mice treated with indicating groups using 30 μg (150 pmol) of BiA9\*2_HF and 3 μg (925 pmol) of E7$_{44-62}$ peptide with and without pTag in the TC-1 model. Tumor growth: E7$_{44-62}$ $n = 6$, BiA9\*2_HF $n = 6$, BiA9\*2_HF + E7$_{44-62}$ $n = 6$, BiA9\*2_HF + pTag$_{9aa}$-E7$_{44-62}$ $n = 8$, tgCD40 from Biocytogen. *$p = 0.0112$, calculated with the Kruskal-Wallis test with Dunn's multiple comparisons test. Survival analysis: E7$_{44-62}$ $n = 6$, BiA9\*2_HF $n = 5$, BiA9\*2_HF + E7$_{44-62}$ $n = 5$, BiA9\*2_HF + pTag$_{9aa}$-E7$_{44-62}$ $n = 7$, tgCD40 from Biocytogen. **$p = 0.0018$ calculated with Kaplan-Meier survival analysis with log-rank test. **B** Survival analysis of mice over time treated in the indicating groups using 30 μg of BiA9\*2_HF and 1.5 μg (450 pmol) of E7$_{44-62}$ peptide linked to the pTag injected s.c or 100 μg intravenously infused selicrelumab. Vehicle $n = 4$, selicrelumab $n = 3$, BiA9\*2_HF + pTag$_{9aa}$-E7$_{44-62}$ $n = 7$, tgCD40 from Biocytogen. *$p = 0.0296$ calculated with Kaplan-Meier survival analysis with log-rank test. **C** Tetramer positive antigen-

specific CD8$^+$ T cells in circulating blood of treated mice performed at day 17. Vehicle $n = 5$, selicrelumab $n = 3$, BiA9\*2_HF + pTag9aa-E744-62 $n = 4$. The whiskers show min and max values, and the box extends from 25 to 75 percentiles, with the middle line representing the median. *$p = 0.0488$ and calculated with one-way ANOVA with Dunnett's multiple comparison test. **D** AST levels in plasma 24 h post first treatment with indicated groups, same mol ratio between the i.v doses. **D** Vehicle $n = 5$, selicrelumab $n = 5$, BiA9\*2_HF i.v $n = 5$, BiA9\*2_HF + pTag$_{9aa}$-E7$_{44-62}$ s.c $n = 5$, tgCD40 from Biocytogen. *P*-values are shown in the graph and calculated with the Kruskal-Wallis test with Dunn's multiple comparison test. **E** Platelet counts after 4 h in ID. Flow. Graph shows fold change of platelets (PLTs), presented as percentage change as compared to test buffer alone. BiA9\*2_HF+pTag$_{9aa}$-KRAS$_{G12D}$ $n = 10$ and selicrelumab $n = 10$. **$p = 0.0021$ calculated with unpaired Mann-Whitney *t* test. **F**, **G** Quantified levels of ALT and ALP in plasma of immunized NHPs with BiA9\*2_HF and pTag$_{9aa}$-KRAS$_{G12V}$ at indicated time-points. Arrows indicate the time of vaccination, and * indicates that only NHP-II got the third treatment.

## Discussion

Precision immunotherapy requires efficient production methods for a fully tailored drug approach. We earlier conceptualized an affinity-based loading technology for rapid peptide cargo exchange onto CD40 agonistic antibodies and coined the technology ADAC. The ADAC technology allows potent CD40-induced immune modulation by flexible peptide cargo delivery to skew the lymph node and tumor microenvironment toward a potent Th1 response. In our first publication, we profiled several existing CD40 agonistic antibodies in the ADAC format and studied the local impact on the lymphoid cells upon transfer of human CD40-expressing cells and therapeutics, along with the impact that the conjugate technology had on the half-life of the

cargo. Herein, we have developed a novel CD40 agonist formatted into the ADAC design and assessed the lead drug compounds impact on T cell expansion, toxicity, and tumor growth in various model systems. The novel anti-CD40 clone, A9, displayed agonistic properties on both moDCs and B cells. The identified novel anti-CD40 antibody clone A9 exhibited epitope binding on the distal CRD1 region of the receptor and showed no competition with CD40L. The agonistic activity occurs in an FcγR independent manner in an IgG2 format, in line with previous work showing the agonistic activity of mAbs targeting this domain[11]. A hybrid subclass was generated based on the IgG1 scaffold with the IgG2 hinge region grafted into mAb architecture. The A9\*1/2 hybrid exhibited higher agonistic activity than its wild-type IgG1 analog (A9\*1),

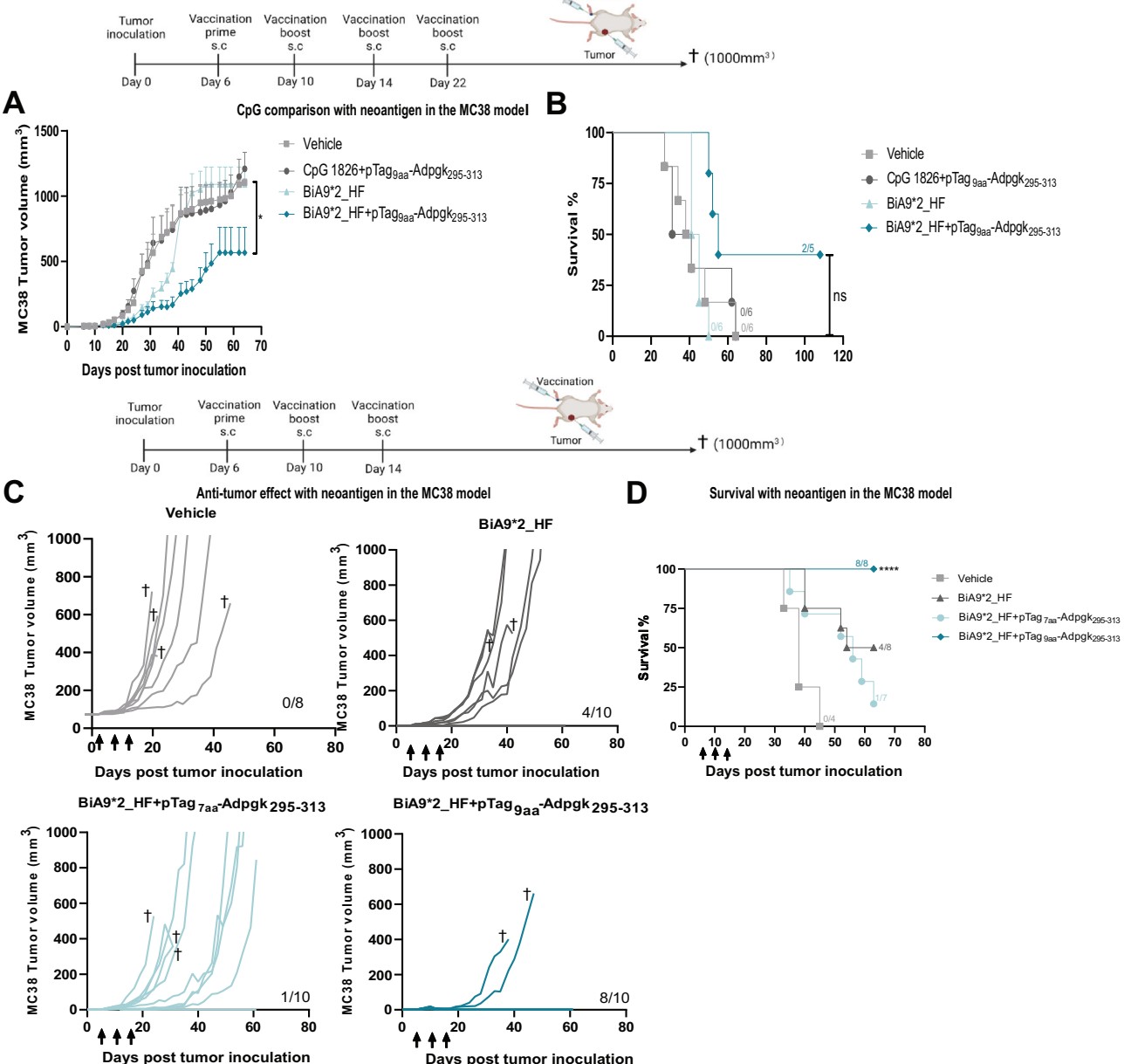

**Fig. 8 | BiA9\*2_HF loaded with tagged neoantigen from an experimental colorectal model leads to superior anti-tumor responses compared to CpG as adjuvant and displays complete tumor regression upon dose elevation.**
**A, B** Tumor volumes over time and survival analysis in mice treated with indicating groups using 30 μg of BiA9\*2_HF and 1.44 μg (450pmol) of pTag$_{9aa}$-Apgk$_{295-313}$ peptide or 20 μg of CpG with 6.4ug (2000pmol) of pTag$_{9aa}$-Adpgk$_{295-313}$. **A** Vehicle $n = 8$, CpG + pTag$_{9aa}$-Adpgk$_{295-313}$ $n = 8$, BiA9\*2_HF $n = 8$ and BiA9\*2_HF + pTag$_{9aa}$-Adpgk$_{295-313}$ $n = 8$, in Biocytogen mice. \*$p = 0.0397$ calculated with the Kruskal-Wallis test with Dunn's multiple comparison test. **B** Vehicle $n = 6$, CpG + pTag$_{9aa}$-Adpgk$_{295-313}$ $n = 6$, BiA9\*2_HF $n = 6$ and BiA9\*2_HF + pTag$_{9aa}$-Adpgk$_{295-313}$ $n = 5$. ns = non-significant calculated with Kaplan-Meier survival analysis with log-rank

test. **C** Individual tumor growth over time in the MC38 model with arrows indicating therapy administration. Vehicle $n = 8$, BiA9\*2_HF $n = 8$, BiA9\*2_HF + pTag$_{7aa}$-Adpgk$_{295-313}$ $n = 10$ and BiA9\*2_HF + pTag$_{9aa}$- Adpgk$_{295-313}$ $n = 10$, tgCD40 from Biocytogen. **D** Survival analysis of mice treated with 30 μg BiA9\*2_HF at the first and third injection and 90 μg at the second injection, either alone or in combination with 1.44 μg (450pmol) pTag$_{9aa}$-Adpgk$_{295-313}$ or 1.32 μg (450pmol) pTag$_{7aa}$-Adpgk$_{295-313}$. At day 14, the vehicle group received 30 μg BiA9\*2_HF with 1.44 μg (450pmol) pTag$_{9aa}$-Adpgk$_{295-313}$. Vehicle $n = 4$, BiA9\*2_HF $n = 8$, BiA9\*2_HF + pTag$_{7aa}$- Adpgk$_{295-313}$ $n = 7$ and BiA9\*2_HF + pTag$_{9aa}$- Adpgk$_{295-313}$ $n = 8$. \*\*\*\*$p = < 0.0001$ calculated with Kaplan-Meier survival analysis with log-rank test. Data is shown as mean with ± SEM (**A**).

though not to the same extent as the IgG2 (A9\*2) based variant. Our observations in the comparison of the agonistic profile of the three IgG isotype generated with the A9 clone are in line with previously reported work in relation to hinge rigidity and impact on initiating CD40 receptor clustering to elicit agonistic activity on APCs[27,30].

The modularity of the ADAC platform is affinity-driven through interaction between the static pTag sequence and the anti-pTag scFv fused to the parental anti-CD40 architecture. In contrast to mRNA-based vaccination strategies, the ADAC platform holds an advantage in

the relative ease in manufacturability of pTag-antigen peptide variants versus mRNA adapted for neoantigen therapeutic use. Further enforcing the possibility of a modular and fast responsive yet cost-effective personalized vaccination pipeline[5,31] and along with the long half-life of the CD40 agonistic antibody, the systemic CD40-induced Th1-focused immunological remodeling itself holds a differentiation factor to vaccines in general. In addition, the affinity binding of the pTag and scFv improve peptide stability, without additional modifications to the peptide, generating a stable drug component ensuring prolonged

exposure time to the peptide therapeutic compared to untagged, or naked peptide therapeutics[25].

The anti-pTag scFv was formatted from a murine antibody and retained the binding to the original pTag$_{18aa}$ sequence after humanization. Intriguingly the humanization significantly increased yields and protein quality post purification compared to its mouse analog. The relative high sequence similarity between the VL regions of the different species framework and careful considerations taken into favorable VH and VL pairing in the humanization process serve as probable contributing factors to the improved quality.

BiAbs are notoriously hampered by developability challenges often related to yield, stability, and aggregation affecting overall functionality[25,32]. The fundamental design of the ADAC molecule is based on the nomenclatural characteristic format for symmetric IgG-scFv bispecific antibodies (2 + 2). Considering the importance of the hinge region in conjuncture with the impact of avidity effects by a bivalent interaction of the CD40 binding arms for optimal agonistic activity, an asymmetric monovalent bispecific design would be unfavorable (1 + 1). Furthermore, with respect to the intended dual mode of action of the molecule, the tetravalent bispecific format works in synergy by providing two pTag binding arms leading to a homogenous Drug to Antibody ratio (DAR) of 2, believed to favorably help promote a robust and homogenous drug conjugate formation with little to no impact on protein quality via the affinity loading strategy. Herein, we screened a set of alternative formats, maintaining a focus on the tetravalent design and thus focused on IgG isotype, fusion point of scFv to parental mAb structure, and choice of linker. Generally, the generated BiAbs based on the fusion of the scFv binding modality to the heavy chain exhibited a superior developability profile compared to the light chain fusion constructs, potentially due to a more favorable folding process in the endoplasmic reticulum (ER) machinery. Moreover, the improved quality and yield of the humanized scFv compared to its murine version was translated when formatted into the ADAC BiAb format. Proposed to be thanks to the overall humanness of the sequence and compatibility of scFv VH-VL frameworks facilitating proper folding.

The longer original tag sequence is derived from tetanus toxoid[33] to ensure that the tag itself does not harbor any human sequence similarities and to ensure safety as the sequence has been part of earlier drug exposure via the national vaccination program. To ensure optimal pTag length for production purposes and to reduce the pTag foreignness, trimming the pTag sequence and its impact on the binding ability to the BiAb was assessed. Trimming to a pTag$_{9aa}$ length did not hamper binding to the BiAb. Moreover, the trimmed pTag did not have any inherent capability to induce T cell responses. The trimmed pTag$_{9aa}$ was confirmed to induce significantly higher CD8$^+$ T cell proliferation in the draining lymph nodes and spleen compared to the original pTag$_{18aa}$ sequence as well as pTag$_{10aa}$ and pTag$_{11aa}$ in vivo when linked to antigen peptide, which could be a result of improved conjugate stability over time when using the shorter tag fragment. In addition, we noted that the linkage of BiAb and pTag-antigen aided antigen-specific CD4$^+$ T cell proliferation, supporting the benefit of using CD40 agonistic antibodies for helper T cell expansion on top of the helper epitope-induced stimulation. We have previously reported that the affinity interaction of ADAC protects the peptide from degradation and prolongs its half-life possibly improving peptide exposure and thereby immunogenicity responses[25]. Herein, our data support that non-linked peptides, together with the BiAb, had a higher percentage of proliferating cells in the popliteal lymph node located closer to the injection site compared to the inguinal lymph node indicating peptide degradation of naked peptides. Conversely, the linked peptide had equal percentages of proliferating CD8$^+$ and CD4$^+$ T cells in the popliteal and inguinal lymph nodes, illustrating the impact of a possible improved peptide stability by ADAC, aiding bioavailability and systemic exposure of peptide-based therapeutics.

The effect of peptide stability was further substantiated when comparing T cell expansion of ADAC with the traditional adjuvant CpG. Using local s.c administration with low dose BiAb together with the pTag-peptide induced T cell proliferation in local and distal lymph nodes on par with CpG together with 12 times higher peptide amounts, while the CpG group with an equivalent peptide amount as with the BiAb did not induce T cell expansion. Conclusively, the data illustrates the ADAC technology's ability, with a low peptide amount and by local administration, to successfully induce both CD8$^+$ and CD4$^+$ T cell activation in vivo, which can decrease the dose and minimize the risk of systemic toxicity by route of administration, compared to traditional systemic drug exposure. Furthermore, ADAC may support inducing immune responses when it comes to ensuring the bioavailability of peptides with poor stability, consequently ensuring increased exposure of any given peptide and, as such, avoiding that drug exposure to the lymph nodes are dictated only by half-life and MHC affinity[34].

Monotherapy of agonistic CD40 antibodies rely on antigen shedding from the tumor and has proven to be effective with high dose i.v or low dose i.p but not low dose s.c at the non-tumor side[18]. The ADAC technology was assessed by administrating a therapeutic conjugated dose s.c on the opposite side of tumor engraftment to minimize direct CD40-mediated anti-tumor effect e.g., mainly focusing on an abscopal effect of the vaccination setting. Importantly, we noted that only when peptide cargo was formulated to allow for conjugate formation, tumor growth control was achieved, which was confirmed in two different tumor models. In the MC38 model, the ADAC formulation managed to transform a previously ineffective but earlier verified neoantigen, into an effective one with superior anti-tumor effect compared to treatment with the neoantigen in conjunction with CpG as adjuvant. Given that the lead antibody clone did not show any signs of induced liver toxicity, we also decided to test if increasing the dose of the antibody/peptide conjugate at the boost injection had an impact on tumor control. Interestingly, we observed an increase in tumor clearance compared to the lower doses tested earlier, which could be due to a combination of induction of effective T cell responses and CD40-induced Th1 remodeling in lymph nodes and tumor compartments. The exact mechanistic explanation and optimal dosing schedule require further investigations ahead. It has previously been illustrated that the CD40 expression is transient and upregulated upon CD40 stimulation by anti-CD40 antibodies in a dose frequency matter followed by DC infiltration in the LNs[18].

In conclusion, any multi-specific affinity protein comprising the anti-scFv pTag:pTag interaction allows for the adaption of the ADAC concept to a multitude of indications. The fully synthetic production route of the individualized part of the drug cargo and the ease of cargo loading via a final in-hospital mixing step is of essence, to avoid biological production steps that other precision immunotherapy approaches make use of. The flexibility of the platform allows for tailor-made pTag-peptide formulations based on patient-specific tumor antigens or alternative cargos. In parallel, there is a continuous miniaturization and optimization of sequencing platforms combined with powerful bioinformatic algorithms and tools to identify neoantigens from biopsy's on-site at hospitals. Collectively, these technological advancements, combined with the ADAC platform, will further strengthen the path to a robust yet flexible personalized cancer therapy infrastructure.

## Methods
### Mice
PMEL-1 mice (Cg-Thy1a/Cy Tg (TcraTcrb)8Rest/J, stock# 005023), a transgenic strain expressing the TCR specific for the MHC-I/hgp100$_{25-33}$ (KVPRNQDWL), were obtained from the Jackson Laboratory[35,36]. The OT-II strain (C57BL/6-Tg-(TcraTcrb)425Cbn/Crl, stock# 004194), expressing the TCR specific for MHC-II/OVA$_{323-339}$

## Table 2 | mAb and bispecific antibody construct index

| Construct index overview | | | | |
|---|---|---|---|---|
| Name | Antibody | Isotype | Position of scFv | Linker |
| A9*2 | mAb | IgG2 | – | – |
| BiA9*2_HF | BiAb | IgG2 | CH3 | FLEX (G4S)2 |
| BiA9*2_HR | BiAb | IgG2 | CH3 | RIGID (EA3K)2 |
| BiA9*2_LF | BiAb | IgG2 | CL | FLEX (G4S)2 |
| BiA9*2_LR | BiAb | IgG2 | CL | RIGID (EA3K)2 |
| A9*1/2 | mAb | IgG1/2 HYBRID | – | – |
| BiA9*1/2_HF | BiAb | IgG1/2 HYBRID | CH3 | FLEX (G4S)2 |
| BiA9*1/2_HR | BiAb | IgG1/2 HYBRID | CH3 | RIGID (EA3K)2 |
| A9*1 | mAb | IgG1 | – | – |
| BiA9*1_HF | BiAb | IgG1 | CH3 | FLEX (G4S)2 |
| BiA9*1_HR | BiAb | IgG1 | CH3 | RIGID (EA3K)2 |
| BiA9*1_LF | BiAb | IgG1 | CL | FLEX (G4S)2 |
| BiA9*1_LR | BiAb | IgG1 | CL | RIGID (EA3K)2 |
| BiA9*2_HF_mscFv | BiAb | IgG2 | CH3 | FLEX (G4S)2 |

Bi = Bispecific, A9 = CD40 clone, *X = IgG subclass, _YZ = Y = position of the scFv (H = CH3, L = CL), Z = type of linker (F = flexible, R = Rigid).

(ISQAVHAAHAEINEAGR), was obtained from the Jackson Laboratory. Only female mice were used of the PMEL-1 and OT-II strains for adoptively transferred cells to reduce the risk of self-antigen recognition linked to the y-chromosome. Two different transgenic human CD40 (htgCD40) mice were used. In both strains, the human CD40 is expressed on both B cells and DCs and lacks endogenous murine CD40 expression. One of the strains was a kind gift from the University of Southampton and previously described by C. Ahonen et al. 2002[37], the second strain is a commercial strain from Biocytogen (C57BL/6-Cd40tm1(CD40)/Bcgen, stock# 110009). All strains are housed and bred together in the animal facility (barrier and specific pathogen-free facility following the formal requirement for animal housing relating to light, temperature, ventilation, and cage size following the regulations stated in SJVFS 2019:9 chapters 16 and 17). controlled environmental conditions) of Uppsala University, Sweden, and follows the obtained ethical permit Dnr: 5.8.18 02686/2019 and Dnr: 5.8.18-18415/2023 approved by the Uppsala regional ethical committee. The ethical approval allowed tumor growth up to 1000 mm³ measured with an electronic caliper and volumes calculated according to the ellipsoid formula. If the mouse on the day of measurement had exceeded 1000 mm³ it was immediately euthanized in compliance with our ethical permit. All mice were euthanized by cervical dislocation.

For in vivo experiments, harvested spleens and inguinal lymph nodes from PMEL-1 and OT-II mice (13–18 weeks old) were made into a single cell suspension by mechanical disruption followed by passing the cells through a 70 μm strainer with a subsequent washing step using RPMI media with 5 mM EDTA (VWR). When required red blood cells (RBC) were lysed by RBC lysis buffer (eBioscience). Cells from the spleen and lymph nodes were mixed and stained with carboxyfluorescein succinimidyl ester (CFSE) (Fisher Scientific Cat: 10013162) according to the manufacturer's protocol, and $10 \times 10^6$ cells were injected into the tail vein of the htgCD40 mice. The indicated therapy doses were given as a flat dose per animal.

The peptides are customized orders from Thermo Fisher Scientific (UK) or Schafer-N (Denmark) with >95% purity. Lyophilized peptides were stored at − 20 °C, and diluted peptides were stored at − 80 °C.

## Phage display selections of novel anti-CD40 scFv binders

Two human synthetic scFv libraries, similar in design and construction to previously described[38], were employed for phage display selection of novel anti-CD40 binders. Briefly, human germline genes IGHV3-23 and IGKV1- 39 were used as library scaffolds, and Kunkel mutagenesis was used to introduce diversity into four of the CDRs (HCDR1-3 and LCDR3). The selection was performed utilizing streptavidin-coated magnetic beads (Thermo Fischer Scientific, Cat: #11206D) and biotinylated avi-tagged human CD40 (hCD40) (AcroBiosystems Cat: CD0-H82E8) as antigen and a procedure basically as described[39], but with a few modifications. The selection pressure gradually increased from one round to the other by decreasing the antigen concentration (327 nM in round 1 to 35 nM in round 4) and by increasing number of washes (five in round 1 and nine in round 4). Amplification of phages between rounds was performed overnight by infection of XL1-Blue E. coli, either on agar plates at 37 °C (rounds 1 and 2) or in solution at 30 °C (rounds 3–4) and M13K07 (New England Biolabs, Cat: N0315S) was used as helper phage. Precipitation of amplified phages was performed using PEG/NaCl, and the pellets were resuspended in a selection buffer (2% BSA, 0.05% Tween-20 in PBS) and used for the next round of selection. Phagemid DNA from selection rounds 3 and 4 were re-cloned to a vector providing the OmpA signal peptide for secretion of the scFv along with a triple-FLAG tag and a hexahistidine (His6) tag at the C-terminus as described[39]. The constructs were subsequently transformed into TOP10 E. coli. 940 clones were expressed and binding to hCD40 could be confirmed by standard ELISA procedure[39] for 159 of these, of which 59 turned out to be unique.

## Anti-pTag scFv humanization and ADAC construct design

An unstructured universal B cell epitope (18aa) derived from tetanus toxoid was used to generate high-affinity antibodies to the same (via immunization and hybridoma generation) and to identify the epitope to which these antibodies bound. Specifically, the humanized scFv was generated by reformatting the identified murine mAb derived from the hybridoma described in earlier work from our group[38]. The sequencing of the hybridomas and assessment of binding of the murine scFv (derived from the hybridoma) were assessed in our previous study[25]. Herein, the murine single chain was humanized by grafting CDRs into human VH and VL germline frameworks with the highest homology to the murine framework as assessed by utilizing the IMGT human germline data base. The IGHV1-46 germline gene exhibited 59.4% sequence similarity and was chosen for grafting of VH CDR regions. In addition to sequence similarity, favorable human VH/VL germline pairing from a biophysical property standpoint was taken into consideration upon choice of VL germline framework resulting in the grafting of VL CDR regions into IGKV1-39 s[40]. To address a putative deamidation site (asn-gly, NG) located in the HCDR2 region, five additional mutated candidates based on the newly humanized scaffold were generated. Thus, the procedure resulted in a total of six CDR-graft variants, denoted NG, DG, GG, NA, QG, and SG.

Eleven anti-CD40 scFv clones isolated from the phage display campaign were reformatted into IgG2 by cloning the VH and VL regions with IgG2 constant regions. Analogously, an IgG1-based antibody of the A9 clone was generated by grafting VH and VL of the IgG1 framework. IgG1/2 hybrid was designed by grafting the hinge region of IgG2 into the IgG1 scaffold. The bispecific constructs of the different A9-based mAbs were generated by anti-pTag scFv fused either to the CL or CH3 domain of the parental antibody architecture. All gene fragments were ordered and cloned into an in-house bicistronic vector designed for mammalian cell antibody production (IDT Technologies). Two sets of linkers were employed linking the scFv to the antibody fusion point, a flexible $(G_4S)_2$ or the stalk based alpha helical rigid

linker (EA$_3$K)$_2$. For detailed naming schematics of constructs, see Table 2. For amino acid and oligonucleotide sequences of the generated constructs, please see the Supplementary construct sequence tables file.

## Antibody production

For initial small-scale expression, IgG and bispecific antibodies, Expi-HEK293 (Thermo Fischer Scientific, Cat: A14527) cells were transfected in 24 deep well cultures. Five days post-transfection, supernatants were purified via protein A magnetic bead-based purification utilizing a KingFisher Flex instrument (Thermo Fischer Scientific). Following elution with 0.1 M glycine pH 2.7, samples were neutralized by the addition of 1 M Tris-HCl, pH 8.8, and buffer exchanged into PBS.

For determination of titer levels of the bispecific constructs, each construct was transfected with the ExpiCHO™ transfection system in 24 deep well plates performed in duplicates with a total 2 µg plasmid DNA in 2.5 ml culture volumes (Thermo Fischer Scientific, Cat: A29133). IgG concentrations in the supernatant were determined by bio-layer interferometry (BLI) measurements in an Octet® RED96e system (Fortébio Biologics by Molecular Devices) with Dip and Read™ Protein A biosensors (Fortébio Biologics by Molecular Devices, Cat: 18-5010) according to the manufacturer's instructions. The supernatant samples from day 5 were diluted 1:1 in 20 mM citric acid pH 4.0, 0.1% BSA (w/v), 0.1% Tween-20, 0.5 M NaCl. A standard curve was prepared from an IgG with the respective concentrations of 700-1 µg/ml.

For the scaled-up production, ExpiCHO™ cells were transfected as suggested by the manufacturer with a total of 0.8 µg/ml plasmid DNA per transfection of respective construct in 25 ml culture volumes and subsequently with selected lead candidates in 400 ml. Expressed IgG and bispecific antibodies were purified by protein A facilitated purification on an ÄktaSTART system (Cytiva) using mAbSelect SuRe columns (Cytiva, Cat: 11003494). A 20 mM sodium phosphate, 0.15 M sodium chloride (pH 7.3) buffer was used as binding and wash buffer, 0.1 M glycine (pH 2.5) as elution buffer, and 1 M Tris–HCl (pH 8.5) as neutralization buffer. Endotoxin levels were measured with LAL Cartridges and Endosafe Nextgen-PTS system (Charles River Laboratories, Cat: PTS2005F) according to the manufacturer's instructions. Larger batch scale production of the final bispecific antibody (BiA9*2_HF) was outsourced to Evitria AG (Switzerland).

## SDS-PAGE

A total of 4 µg of the purified IgG samples were mixed with 3x loading buffer (0.1 M Tris–HCl, 45% glycerol, 0.03% bromophenol blue, 0.3% SDS) for non-reducing conditions and for the reducing analysis mixed with 3x times loading buffer containing 0.15 M Tris 2-carboxyethyl-phosphine hydrochloride and incubation at 95 °C for 7 min. The samples were run on a 4–20% Criterion™ TGX Stain-Free™ protein gel (Bio-Rad Laboratories, Cat: 5678094) according to the company's protocol. The bands were visualized by staining the gel in GelCode™ Blue Safe protein stain (Thermo Fisher Scientific, Cat: 24594) for 1 h at room temperature and gentle shaking.

## Size exclusion chromatography (SEC)

In total, 25 µg IgG in 100 µl was injected onto a Superdex Increase 200 10/30 GL gel filtration column (Cytivia, Cat: 28990944) coupled to an Agilent 1200 series HPLC system (Agilent Technologies). SEC runs were performed at a 0.5 ml/min flow rate with PBS as a running buffer. Eluted protein fragments were detected by an online 280 nm absorption measurement.

## Surface plasma resonance (SPR)

Kinetic measurements of scFv fragments, IgGs, and bispecific constructs were performed by surface plasmon resonance (SPR) on a Biacore T200 instrument (GE Healthcare). For analyses of the scFvs, an anti-FLAG M2 antibody (Sigma-Aldrich, Cat: F3165), functioning as a capture ligand, was immobilized onto all 4 surfaces of a CM5-S amine sensor chip (Cytiva, Cat: 29149603) according to manufacturer's recommendations. Analogously, for analyses of IgGs, an anti-Fab antibody (Cytiva, Cat: 28958325) was used as a capture ligand. A kinetic screen of the 59 sequence-unique scFvs was performed by injecting these, allowing capture on the chip surface, followed by injection of 50 nM hCD40-Fc (R&D Systems, Cat: 1493-CD). Affinity measurements to the pTag and its trimmed derivates by the anti-pTag murine scFv and humanized versions were performed using the same approach. Following capture, a 5–fold dilution series comprising five concentrations ranging from 0.16–100 nM of the peptides was injected over the chip. Purified anti-CD40 IgG2 mAb clones were captured on the anti-Fab chip with equal response units. A 3-fold dilution series of hCD40-Fc (R&D Systems), 4-fold dilution series of hCD40-Avi and cynomolgus monkey (Cyn) CD40-Fc (R&D Systems, Cat: 9660-CD) were injected, ranging from 1.2–100 nM and 0.31–80 nM. All experiments were performed at 25 °C in a running buffer (HBS supplemented with 0.05% Tween20, pH 7.5). As a regeneration buffer, 10 mM glycine HCL pH 2.1 was used. By subtracting the response to a reference surface, being an a-FLAG M2 or anti-Fab antibody immobilized surface, response curve sensorgrams for all clones were obtained. Data was analyzed using the Biacore T200 Evaluation 3.1 software, and kinetic parameters were calculated assuming a 1:1 Langmuir binding model. Purified bispecific variants were immobilized on a CM5 amine sensor chip (Cytiva, Cat: 29149603) by amine coupling on the Biacore 8 K instrument (Cytiva). The analyte hCD40-Fc was injected in a concentrations series of 6 starting at 100 nM with a dilution factor of 2 in PBST or a start concentration of 1000 nM for peptides. The data was analyzed using the BIA8K Evaluation software with a 1:1 fitting model for the determination of Kd.

## CD40L blocking by SPR

CD40 ligand (CD40L) blocking by SPR (Biacore T200) was assessed by immobilizing an anti-human Fab antibody as described above. The eleven tested anti-CD40 IgG2 antibodies were individually injected and captured on the chip surface. Human CD40-Fc was diluted to 20 nM in HBS supplemented with 0.05% Tween-20 injected either alone or pre-incubated with 200 nM human CD40L (R&D Systems, Cat. 6420-CLB) and subsequently added. As a negative control, 200 nM of solely CD40L was injected. Following the dissociation phase, the chip surfaces were regenerated with 10 mM glycine-HCl with pH 2.1. Data was analyzed using BIAeval v.3.1.

## Antibody clones Off-target binding screening

Potential off-target binding was tested by ELISA against baculovirus particles (BVP), similar to described[41], and against a panel of antigens; Cardiolipin (Sigma-Aldrich), Keyhole Limpet Haemocyanin (KLH) (Sigma-Aldrich), lipopolysaccharide (LPS) (InvivoGen), single-stranded DNA (ssDNA) and double-stranded DNA (dsDNA) from calf thymus (Sigma-Aldrich) and human insulin (Sigma-Aldrich) analogous to previous described[42]. The BVP was produced in-house by PEG/NaCl precipitation from the supernatant of baculovirus-infected SF9 cells. In the ELISA, the BVPs (1:20), Cardiolipin (50 µg/ml), KLH (5 µg/ml), LPS (10 µg/ml), ssDNA (1 µg/ml), dsDNA (1 µg/ml) and insulin (5 µg/ml) were diluted in PBS and added to the wells of an ELISA plate and incubated overnight at 4 °C. The plates were washed and blocked with PBS 0.5% BSA for 1 h at room temperature (RT). The eleven anti-CD40 IgG2 antibodies were diluted to 100 nM and 20 nM in PBS containing 0.5% BSA added to the plates and incubated for 1 h at room temperature. The secondary goat anti-human IgG kappa-HRP (Southern Biotech, Cat. 2060-05) was added and incubated for 1 h prior to TMB (Fisher Scientific, Cat. N301) development and stopped with 1 M H$_2$SO$_4$ and read at 450 nm.

For the final BiAb drug candidate, an array-bases screening for off-target binding was tested using Retrogenix Cell Microarray

Technology with live and fixed cells, including 6105 full-length human plasma membrane proteins, secreted and cell surface-tethered human secreted proteins plus a further 400 human heterodimers, performed by Charles River Laboratories using 5 μg/ml of antibody.

## HDX-MS Epitope mapping

Human CD40-Fc (R&D Systems) was diluted to 0.5 mg/ml in PBS and mixed with a 1:1 molar ratio of the A9 or B1 antibody. The complexes were concentrated by a 10 KDa cut-off centrifugation filter (Amicon Ultra-0.5, Merck). All liquid handling for labeling and quenching was done using an automated HDX MS system (CTC PAL/Biomotif HDX). The samples were labeled for 10 min, 25 min, and 60 min and initiated by the addition of deuterated PBS. The reaction was stopped by decreasing the pH to 2.3 and temperature to 4 °C by the addition of a quenching solution containing 6 M Urea, 100 mM TCEP, and 0.5% TFA. Each labeled and quenched sample was analyzed in an automated HDX MS system (CTC PAL/Biomotif HDX) in which samples were automatically labeled, quenched, digested, cleaned, and separated at 2 °C. Samples were digested using an immobilized pepsin column (2.1 × 30 mm) at 250 μL/min, followed by an online desalting step using a 2 mm I.D x 10 mm length C-18 pre-column (ACE HPLC Columns, Aberdeen, UK) using 0.05% TFA at 350 μl/min for 3 min. Peptic peptides were then separated by a 15 min 8-60 % linear gradient of ACN in 0.1% formic acid using a 2 mm I.D x 50 mm length HALO C18/1.8 μm analytical column operated at 70 μL/min. An Orbitrap Q Exactive mass spectrometer (Thermo Fisher Scientific) operated at 70,000 resolution at m/z 400 was used for analysis. The mascot was used for peptide identification. HDExaminer software (Sierra Analytics, USA) was used to process all HDX-MS data. Differential deuteration uptake compared between the CD40 receptor alone or in complex with mAb clone A9 or B1 was calculated. Kinetics for 45 peptides covering 55% of the protein construct were evaluated.

## Deimmunization of the CD40 antibody

To assess potential immunogenicity risks, screening for potential Th-cell epitopes was performed using the SciCross Immunogenicity Platform (SCIP). The T cell epitope prediction algorithms of SCIP are based on statistics and machine learning, including support vector machines and optimization techniques[43,44]. The HLA class II alleles covered in the analysis give a high global population coverage. A region with a relatively high immunogenicity risk was identified in the FR2-CDR2 junction of the scFv sequence. SCIP was then used to identify amino acid changes in this region that reduce the immunogenicity risk, by means of reducing the number of predicted T cell epitopes. This was done through a combinatorial search of variants by introducing single, double, or triple amino acid changes. A set of potential sequence modifications was identified, also considering experience in amino acid substitutions acceptable to keep target binding. The S/F variant shows the same affinity as the original variant, as illustrated in supplementary materials (Supplementary Fig. 3).

## Flow cytometry

For the flow cytometry analysis all antibodies were purchased from Biolegend, Abcam or BD Biosciences, the following anti-human antibodies were used; anti-HLA-DR (L243, PE-Cy5), anti-CD83 (HB15e, APC), anti-CD86 (IT2.2/BU63, BV421/PE), anti-CD14 (63D3, APC-Cy7), anti-CD1a (HI149, PE), anti-CD19 (HIB19/CB19, APC-Cy7/PE) anti-CD20 (2H7, FITC) and anti-HLA-A3 (GAP.A3, PE). The anti-mouse antibodies used; anti-CD11c (N418, PE), anti-CD11b (M1/70, PerCP), anti-I-A/I-E (M5/114.15.2, APC), anti-CD86 (GL-1, BV421), anti-CD3 (17A2, BV421/APC-Cy7), anti-CD4 (RM4-4, PerCP-Cy5.5), anti-CD8 (53-6.7, APC/BV785), anti-CD90.1 (Thy1.1) (OX-7, PE) and anti-H-2D(b) RAHYNIVTF tetramer (PE) produced by NIH Tetramer Core Facility[4]. A detailed summary of all antibodies used within this study for FACS analysis, including conjugated fluorochrome, clonal information, dilutions, and

catalog numbers, can be found in the Supplementary information file (Supplementary Table 1). Gating strategies for respective FACS experiment and figure can be found in the Supplementary information file (Supplementary Fig. 5). Prior to staining, the cells were washed once with PBS (Fisher Scientific, Cat: 15326239) containing 1% BSA (Sigma, Cat: A9647) and then the staining antibodies were diluted in 50 μl/sample, added to the cells and incubated for 20 min at 4 °C. The final washing was then performed, and the cells were resuspended in PBS containing 3 mM EDTA (VWR, Cat: E177). For tetramer staining whole blood was stained with tetramer and surface antibodies for 30 min at RT in the dark. The red blood cells were lysed using 1X RBC lysis buffer (Thermo Fischer Scientific, Cat: 12770000) and incubated for 5 min at RT, and the reaction was stopped with PBS. The cells were washed and resuspended in PBS containing 3 mM EDTA. The cells were run by using the CytoFlex Flow Cytometer (Beckman Colter Life Science)

## Monocyte & B cell isolation

Buffy coats from voluntary blood donations from the academic hospital in Uppsala were purchased and used for harvesting immune cells. After PBS dilution of the buffy coat, PBMC isolation was performed using the SepMate™ (Stemcell, Cat: 85460) tubes according to the manufacturer's protocol. In brief, Ficoll-Paque (VWR, Cat: 17-5442-03) was added to each tube and the blood suspension was layered carefully onto the Ficoll. The separation was performed by centrifugation at 1200 × g for 10 min at RT. The PBMC fraction was isolated and washed twice with PBS, first centrifuged at 300 × g for 8 min followed by 200 × g for 5 min. Thereafter the specific cell populations were isolated with either the human CD14 microbeads (Miltenyi Biotec, cat: 130-050-201) or the human pan B cell isolation kit (Miltenyi Biotec, cat: 130-101-638) using the LS columns (Miltenyi Biotec, cat: 130-042-401) with the MACS Separator system (Miltenyi Biotec) or alternatively for B cells utilizing a Human untouched B cell Isolation kit (Thermo Fischer Scientific, Cat: 11351D). The purity of the isolation was determined with flow cytometry (Cytoflex, Beckman Colter) by either staining with anti-CD14 or anti-CD19 antibodies in the positive and negative fractions, and a purity above 90% was acceptable.

## Dendritic cell culture and in vitro stimulation

Isolated CD14 + cells were cultured for 6 days with the differentiation medium of Roswell Park Memorial Institut (RPMI) 1640 GlutaMax (Thermo Fischer Scientific, Cat: 61870-010) supplemented with 10% FBS (Fisher Scientific, Cat: 17573595), 1% penicillin/streptomycin (PeSt) (Fisher Scientific, Cat: 11548876), 1% Hepes (Thermo Fischer Scientific, Cat: 15630-056) together with 75 ng/ml human GM-CSF (Peprotech, Cat: 300-03) and 50 ng/ml human IL-4 (Peprotech, Cat: 200-04). The cells were cultured at a density of $1 \times 10^6$ cells/ml at 37 °C in 5% $CO_2$ in tissue-treated (TC) plates. On days 3 and 5 of the culture, half of the medium was exchanged with the addition of the cytokines to the total volume. At day 6 the differentiation of immature DCs (imDC) was evaluated by flow cytometry by assessing the downregulation of the CD14 marker and upregulation of CD1a. The imDCs were antibody-stimulated for either 6 h or 48 h. Post 6 h, the cells were washed twice and incubated for a total of 48 h with the indicated antibody concentrations in the figures and with 1 μg/ml LPS (Sigma, Cat: L4391) as positive control. The activation was evaluated by quantification of surface protein expression using flow cytometry and by measuring IL-12p40 secretion from the collected supernatant.

## Human and murine IL-12p40 detection by ELISA or Mesoscale

Human IL-12p40 was quantified according to the manufacturer's protocol in the supernatant from the DC activations by either human IL-12p40 ELISA (Biolegend, Cat: 430701) or human U-plex IL-12p40 (Mesoscale Discovery, Cat: K151UQK). In brief, ELISA high binding

plates (Sarstedt, Cat: 82.1581.200) were coated with 1:200 dilution of the capture antibody in PBS with 1% BSA and 0.05% Tween-20 (Sigma, Cat: P9416) and incubated overnight at 4 °C. Between each incubation step, the plates were washed 3 times with PBS containing 0.05% Tween-20. The plates were blocked with PBS containing 3% BSA and incubated for 1 h at RT prior to the addition of the diluted supernatant and standard for an additional 2 h incubation at RT. Subsequently, the detection antibody was diluted 1:200 in the same buffer as the capture antibody and incubated for another 1 h at RT prior to the addition of 1:1000 diluted avidin-HRP (Thermo Fischer Scientific, Cat: 434323) incubated for 30 min at RT. The plates were developed with TMB and stopped with 1 M $H_2SO_4$, and absorbance was read at 450 nm.

The murine IL-12p40 ELISA followed the same procedure and buffers as the human IL-12p40 ELISA except that the coating antibody was diluted in 0.05 M carbonate-bicarbonate buffer. For coating, 1 μg/ml of the coating antibody (clone: C15.5, Biolegend) and 1 μg/ml of the detection antibody (clone: C17.8, Biolegend) were used. The avidin-HRP (Dako Cat: P0347) was diluted 1:4000 before addition and incubated for 1 h at RT. The plates were developed with TMB and stopped with 1 M $H_2SO_4$, and absorbance was read at 450 nm.

### AST ELISA
Mouse plasma AST levels were quantified with the mouse AST SimpleStep ELISA (Abcam, Cat: ab263882) according to the manufacture´s protocol. In brief, the plasma samples were diluted between 1:200 – 1:4300 and added to the pre-coated wells together with the antibody cocktail, and incubated for 1 h at RT. Followed by washing with 1x washing buffer. The plates were developed with TMB, and the reaction was stopped by the addition of stop solution, and the absorbance was measured at 450 nM.

### B cell activation assay
The isolated B cells were diluted in RPMI-1640 GlutaMax supplemented with 10% FBS, with or without 1% PeSt and 1% Hepes. The cells were seeded in a 96-well plate at a density of $1.5 \times 10^5$ cells/well and stimulated with the indicated antibodies and concentrations for 24 h at 37 °C in 5% $CO_2$. The CpG Oligodeoxynueclotide 2006 (CpG-ODN 2006) (InvivoGen, Cat: tlrl-2006) with a concentration of 5 μM was used as a positive control. The activation was evaluated by upregulation of CD86 by flow cytometry.

### ELISpot
Cryopreserved PBMCs were thawed in RPMI 1640 GlutaMAX™ medium supplemented with 10% FBS, 1% penicillin-streptomycin, 1% HEPES and 50 U/ml Pierce™ nuclease (Fisher Scientific, Cat: 12391963), washed, resuspended in the same medium (excluding nuclease) and were left to rest for 2 h at 37 ̊C, 5% $CO_2$ at a concentration of $2 \times 10^6$ cells/ml. After resting, cells were harvested and seeded in pre-coated IFN-γ ELISpot plates (Mabtech, Cat: 3420-2APT) at a concentration of $3 \times 10^5$ cells/well to measure IFN-γ secretion, following the manufacturer's protocol. In short, the pre-coated plates were washed and blocked, and each peptide ($pp65_{489-510}$, $pTag_{9aa}$, and $pTag_{9aa}$-$pp65_{489-510}$) was added in triplicates (10 μM final concentration) followed by each donor's cells, and the plates were incubated at 37 ̊C with 5% $CO_2$ for 24 h. For positive control, an anti-CD3 monoclonal antibody was employed (CD3-2, Mabtech) at a final dilution of 1:1000.

For the DC:T cell co-culture and following ELISpot analysis, PBMCs confirmed HLA-A3$^+$ were prepared from fresh buffy coats, and monocytes were isolated as described above. Monocytes were cultured in the complete medium described above, supplemented with GM-CSF (75 ng/ml) and IL-4 (50 ng/ml) for three days. On day 3, imDCs were loaded with $pTag_{9aa}$-$CMVpp65_{489-510}$ peptide (10 μM final concentration) and stimulated with LPS (1 μg/ml final concentration). On the same day, PBMCs of the same donors were thawed in RPMI 1640 GlutaMAX™ medium supplemented with 10% FBS, 1% PeSt, 1% Hepes, and 50 μ/ml Pierce™ nuclease. Next, the cells were washed, and CD8 T cells were isolated according to the manufacturer's instructions (Miltenyi, Cat: 130-096-495). After isolation, CD8 T cells were resuspended in the same medium excluding nuclease and supplemented with 5 ng/ml IL-7 (Peprotech, Cat: 200-07) and transferred to culture dishes at $2-3 \times 10^6$ cells/ml and placed in an incubator at 37 °C in 5% $CO_2$. On day 4, the mature DCs and CD8 T cells were collected and washed. Cells were resuspended in a complete medium and counted, and 60 ng/ml of IL-21 (Peprotech, Cat: 200-21) was added to the CD8 T cells. Next, the DCs were mixed with the CD8 T cells of the same donor, resulting in a 4:1 T cell:DC ratio. The mixed cell suspensions were plated out in a 48-well culture plate, in 500 μl/well, six wells for each donor ($5 \times 10^5$ T cells:1.25x $x10^5$ DCs per well). The plates were placed in an incubator at 37 °C in 5% $CO_2$. On day 7, day 9 and day 11, the new medium supplemented with IL-7 and IL-15 (Peprotech, Cat: 200-15), final concentrations 5 ng/ml, was added to the wells. On day 14, cells were harvested and combined per donor and seeded in pre-coated IFN-γ ELISpot plates at a concentration of $3 \times 10^5$ cells/well to measure IFN-γ secretion, following the manufacturer's protocol. In short, the pre-coated plates were washed and blocked, and $pTag_{9aa}$ peptide was added in triplicate wells (2 μM final concentration) followed by each donor's cells, and the plates were incubated at 37 °C with 5% $CO_2$ for 20 h. For the positive control, an anti-CD3 monoclonal antibody was employed (CD3-2, Mabtech) at a final dilution of 1:1000. The plates in the two above-described set-ups were developed with an alkaline-phosphatase-conjugated detection antibody (7-B6-1-ALP, Mabtech) which was added to the plates for 2 h (1:200), followed by the BCIP/NBT substrate (Mabtech, Cat: 3650-10) which was incubated for 10 min and then washed. The plates were left to dry overnight and were analyzed using a Mabtech IRIS™ reader coupled with Mabtech Apex™ software for spot enumeration. The threshold for a positive peptide-specific T-cell response was a minimum of a three-fold increase in Spot Forming Units (SFU) above its own negative control.

Mouse IFNγ pre-coated ELISpot plates (Mabtech, Cat: 3321-4APT) were used for the detection of IFNγ secretion from isolated lymphocytes from vaccinated mice at day 0, 5, and 10, and isolation occurred at day 14. A total of $3 \times 10^5$ spleen and lymph node cells with a 1:1 ratio per well were stimulated with peptides ($pTag_{9aa}$ or $E7_{44-62}$ CD8 epitope) with 5 μM final concentration for in 37 °C with 5% $CO_2$ for 22 h. Con A (Fisher Scientific, Cat: 15566286) was used as a positive control at a final dilution of 1:500. After removal of the cells, 1 μg/ml biotin conjugated detection antibody (Mabtech, clone: RA-6A2) was added to the wells for 2 h incubation. The plates were developed by incubation with 1:1000 diluted streptavidin-ALP in 0.5% FBS/PBS, followed by the BCIP/NBT substrate addition. The plates were left to dry overnight and were analyzed using the aforementioned Mabtech IRIS™ reader coupled with Mabtech Apex™ software for spot enumeration.

### BMDC isolation and differentiation
The femur and tibia were isolated from the htgCD40 mice in sterile conditions as previously[25]. Briefly, the soft tissue was dissected, and the bones were disinfected in 70% ethanol and washed in sterile PBS before being cut open to flush out the bone marrow (BM) precursor cells with sterile Iscove's Modified Dulbecco's Medium (IMDM) Glutamax (Thermo Fischer Scientific, Cat: 31980-022). The bone marrow cells were cultured for 8 days at 37 °C with 5% $CO_2$ in non-TC treated plates (Gibco) at a concentration of $2.5 \times 10^5$ cells/ml in 20 ml of IMDM Glutamax medium supplemented with 10% FBS, 1% penicillin/streptomycin, 1% Hepes, 50 μM 2-mercaptoethanol (Fisher Scientific, Cat: 31350010) with addition of 20 ng/ml of mGM-CSF (Biolegend, Cat: 576306). On day 3, an additional 20 ml of complete media supplemented with 20 ng/ml of mGM-CSF was added, and on day 6, half of

the media was replaced with fresh complete media supplemented with 20 ng/ml of mGM-CSF. At day 8, the immature bone-marrow dendritic cells (imBMDCs) were harvested, and the differentiation was checked by flow cytometry by dual expression of CD11c and CD11b, then plated $1 \times 10^5$ cells/well and stimulated with the antibodies for 48 h at 37 °C with 5% $CO_2$ prior to harvesting the supernatant for IL-12p40 quantification.

## NHP B cell stimulation

Blood from four cynomolgus monkeys (2 males and 2 females) was purchased (SILABE), and PBMCs were isolated using Ficoll-Paque™ and subsequently frozen down. For the B cell stimulation, frozen cynomolgus PBMCs were thawed and seeded at $0.5 \times 10^6$ cells/well in 100 μl RPMI-1640 + 10% FBS in 96 well TC plate. Next, PBMCs were stimulated with concentration titration of BiA9*2_HF and incubated for 24 h at 37 °C and 5% $CO_2$. Subsequently, cells were harvested and stained with human Fc block (Miltenyi, Cat: 130-059-901) for 30 min, and then the cells were washed and stained with CD86 and CD20 for 1 h. After a final wash, cells were analyzed using flow cytometry.

## NHP in vivo study

The non-human primate (NHP) study was approved by the Stockholm Regional Ethical Board on Animal Experiments (Dnr: 18427-2019). Two female cynomolgus monkeys were housed in the Astrid Fagraeus laboratory at Karolinska Institute, according to the guidelines of the Association for Assessment and Accreditation of Laboratory Animal Care. The monkeys were immunized twice (NHP-I) or three times (NHP-II) with BiA9*2_HF+pTag9aa-KRASG12V at 0.22 mg/kg for NHP-I and 0.16 mg/kg for NHP-2, with 2-weeks interval between the injections. The monkeys were monitored and bled at several time points, and plasma was isolated from the collected blood. Clinical chemistry blood analysis, performed by Scantox, to quantify AST, ALP, and urea nitrogen levels as a measure of liver and kidney function. IL-12/IL-23p40 in the plasma was determined by mesoscale analysis using the V-PLEX Cytokine Panel 1 NHP Kit (Mesoscale Diagnostics, Cat: K15057D). The assay was performed by SciLifeLab, Affinity Proteomics, and Uppsala.

## ID.Flow

The ID.Flow was performed by Immuneed and is briefly described here. Fresh whole blood was taken from 10 healthy volunteers (5 males and 5 females, age > 45) with obtained informed consent and ethical approval from the Uppsala regional ethical committee (Dnr: 2015/325). A low amount of soluble heparin was added to the blood immediately upon collection. The blood was then immediately transferred to the test system, followed by administration of BiA9*2_HF (50 nM) with pTag9aa-KRASG12V (150 nM) or selicrelumab (50 nM), and set to circulate at 37 °C to prevent clotting. Blood was sampled at baseline and from the test system at baseline, 15 min, 1 h, and 4 h, and analyzed using flow cytometry. Platelet counts, identified as CD41 +, CD45-, were measured at 4 h with Sysmex XN-350 Hematology Analyzer. White blood cell (WBC)-Platelet conjugates, identified as CD41 + CD45 +, were measured at 15 min and 1 h.

## In vivo T cell immunogenicity

Adult htgCD40 mice from Biocytogen, 9–13 weeks, both females and males equally distributed in the groups, were injected subcutaneously (s.c) on the right side in the hock with either 25 mM histidine pH 6 buffer as vehicle, 1350 pmol (4.4 μg) pTag9aa-E744-62 peptide alone, 450 pmol (90 μg) BiA9*2_HF antibody alone or combination of various doses of BiA9*2_HF antibody and pTag9aa-E744-62 peptide with 1:3 molar ratio on day 0, day 5 and day 10. On day 14, the draining popliteal, draining inguinal, and spleens were collected. The organs were passed through a 70 μm cell strainer, and in the spleens, the RBC

were lysed by using RBC lysis buffer prior to performing a mIFNγ ELISpot assay.

## T cell in vivo proliferation assay

Adult htgCD40 mice from Biocytogen, 8–14 weeks females were used for the in vivo CD8 T cell proliferation assays and 8–13 weeks females and males from Southampton were used for the in vivo CD4 T cell proliferation assays. CFSE-labelled ($10 \times 10^6$ cells) PMEL-1 (13–18 weeks females) or OT-II (12-16 weeks females) immune cells were injected intravenously (i.v) in the tail vein. The next day, the mice were s.c vaccinated on the right side in the hock with the BiA9*2_HF antibody combined with either the gp100_{20-39} peptide without or with different lengths of the pTag or OVA_{323-339} peptide with and without pTag9aa (Supplementary Table 2). Adult htgCD40 mice from Biocytogen, 12–14 weeks females and males, were used in a two-time vaccination strategy post-PMEL-1 (12-18 week females) immune cell transfer with a follow-up vaccination performed 5 days after the prime vaccination. The next day, the mice were s.c vaccinated on the right side in the hock with the gp100_{20-39} peptide alone, BiA9*2_HF + gp100_{20-39}, BiA9*2_HF + pTag9aa gp100_{20-39} and gp100_{20-39} peptide mixed with CpG-1826 (InvivoGen Cat: tlrl-1826). The draining popliteal, draining inguinal, non-draining inguinal lymph nodes and spleens were collected two days post the last vaccination. The organs were passed through a 70 μm cell strainer, and in the spleens, the RBC were lysed by using 1x RBC lysis buffer prior to surface marker staining and analysis by flow cytometry.

## In vivo anti-tumor efficacy

Adult htgCD40 mice from Biocytogen (7–15 weeks, both females and males) were used to evaluate the ADAC technology's potential to induce an anti-tumor response. The murine lung Tumor Human papillomavirus-16 (HPV-16) E6/E7 TC-1 cell line from John Hopkins was utilized as the tumor model. The TC-1 cell line was cultured in RPMI Glutamax supplemented with 10% FBS, 1% PeSt, 1 mM Sodium pyruvate (Fisher Scientific Cat: 11530396), and 10 mM Hepes with a seeding density of $3 \times 10^4$ cells/cm². The cells were split every 2-3 days with the use of trypsin (Fisher Scientific, Cat: 10779413). Mice were injected with $5 \times 10^4$ TC-1 cells in PBS s.c. in the right flank. At days 5, 10, and 15 post-tumor inoculation, the mice were treated s.c with hock injection on the left hind leg with 30 μg of BiA9*2_HF and 3 μg of pTag9aa-E744-62 peptide, in the peptide alone group 30 μg E744-62 peptide was used (Supplementary Table 2). To generate BiAb/peptide complexes, the two entities were incubated for 1 h at RT, with a 1:3-6 BiAb to peptide molar ratio. In the comparative study with selicrelumab, BiA9*2_HF was injected in the same manner and intervals as described above with 30 μg of the bispecific antibody and 1.5 μg of pTag9aa-E744-62 peptide. For selicrelumab, 100 μg was administered i.v at days 5, 10, and 15. Tetramer staining for antigen-specific T cells was performed on day 17, as described above. Tumor growth in all trials was monitored every 2-3 days until reaching a volume of 1000 mm³, measured with an electronic caliper, and volumes calculated according to the ellipsoid formula. Mice that reached the humane endpoint, including 10% weight loss or ulcerating tumors, were excluded from survival analysis except if the tumor sizes were larger than 800 mm³, then they were included in the survival analysis with an estimated time of death.

A second tumor model was assessed to target neoantigens using the MC38-L cell line, kindly provided by Mario P. Colombo lab. The cells were cultured in IMDM Glutamax supplemented with 10% FBS, 1% PeSt, and 25 μM 2-mercaptoethanol with a seeding density of $2 \times 10^4$ cells/cm² and split every 2-3 days with trypsin. Adult tghCD40 mice from Biocytogen (7–11 weeks females and males) were injected with $3 \times 10^5$ MC38-L cells in PBS s.c on the right flank. At days 6, 10, 14, and 22 post-tumor inoculation, the mice were vaccinated s.c with hock injection in the left hind leg with 30 μg of either; BiA9*2_HF alone or together with 1.5 μg of pTag9aa-Adpgk or the peptide only or 20 μg of

CpG with 6.4 μg pTag$_{9aa}$-Adpgk peptide. In the follow-up study, 30 μg of BiA9*2_HF was used in the first and third injections, and 90 μg used in the second injection, either alone or with 1.5 μg of pTag$_{9aa}$-Adpgk peptide or a truncated tag version pTag$_{7aa}$-Adpgk. A single 90 μg dose of the BiA9*2_HF with pTag$_{9aa}$-Adpgk$_{295-313}$ was administrated at day 19 to the vehicle group to assess if one administration of a high dose could impact tumor growth control. The tumor volumes were monitored and measured as previously described and using the same exclusion criteria.

**Data and statistical analysis**

The flow cytometry data was analyzed with Kaluza Analysis 2.1 or FlowJo. The graphs, peak integrations, and statistics were made with GraphPad Prism 8.0 or 9.0 (GraphPad Software, USA). The heatmap was generated in R (v.4.2.0) using the package ComplexHeatmap (v.2.12.0). Statistical comparisons were performed using Kruskal–Wallis with Dunn's correction for multiple comparisons, one-way ANOVA with Dunnett's correction for multiple comparisons, unpaired Mann-Whitney *t* test, unpaired two-tailed t-test and Kaplan-Meier survival analysis with the log-rank test.

**Reporting summary**

Further information on research design is available in the Nature Portfolio Reporting Summary linked to this article.

## Data availability

Amino acid sequences and oligonucleotide sequences of Antibody-based and Bispecific ADAC constructs are available in Supplementary Data File 1 and Supplementary Data File 2, respectively. Amino acid sequences for peptide variants can be found in the Supplementary Information File and Supplementary Table 2. All oligonucleotide Sequences of generated constructs have been deposited in GenBank under the accession codes [BankIt2871286 A9_L PQ331142, BankIt2871286 A9_LF PQ331143, BankIt2871286 A9_LR PQ331144, BankIt2871286 A9*1_H PQ331145, BankIt2871286 A9*1_HF PQ331146, BankIt2871286 A9*1_HR PQ331147, BankIt2871286 A9*1/2_H PQ331148, BankIt2871286 A9*1/2_HF PQ331149, BankIt2871286 A9*1/2_HR PQ331150, BankIt2871286 A9*2_H PQ331151, BankIt2871286 A9*2_HF PQ331152, BankIt2871286 A9*2_HR PQ331153]. Further information and requests for resources and reagents should be directed to and will be addressed by co-first authors A.M. and I.L. and co-corresponding authors, S.M. and J.R. Source data are provided with this paper.

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

## Acknowledgements

The research has been funded by grants from Swedish innovation agency Vinnova through GeneNova (2021-02640), CellNova (2017-02105), AdBIOPRO (2016-05181), Knut and Alice Wallenberg Foundation (WCPR) (J.R.), Knut and Alice Wallenberg foundation 2020.0182 (S.M.), Cancerfonden 22 2219 Pj (S.M.), SciLifeLaboratory Drug Development and Discovery platform project (S.M.), Svenska Sällskapet för Medicinsk Forskning (SSMF) S15-0065 (S.M.), SweDeliver (Vinnova 2019-00048) and SweLife (S.M.). We would like to acknowledge Professor TC Wu at Johns Hopkins University for sharing the TC-1 cell line and Claudia Chiodoni at Fondazione IRCCS Istituto Nazionale Tumori for sharing the MC38L cell line. Also, Mark Cragg at the University of Southampton for providing the transgenic human CD40 mice together with Dortmund College as the owner of the strain. We also like to acknowledge Tina Furebring and Lindvi Gudmunsdotter for their invaluable insights and input on the herein-presented work. We also thank Karin Lore and Xianglei Yan for their technical support in executing the cyno-molgus study.

## Author contributions

Conceptualization of the idea by S.M. Design of bispecific proteins, generation, characterization, and data analysis (protein production, biophysical properties, kinetics) lead by A.M and assisted by J.P. Selection and initial characterization of antibody by S.M., H.P., A.O., O.A., L.D., and I.L. lead in design and performing in vitro studies, FACS based readouts and data analysis of in vitro and in vivo studies together with co-authors. In vivo studies designed by S.M., G.G.A. I.L. and R.V and carried out by I.L., R.V., G.G.A. and A.K. Data analysis of in vitro studies performed by M.L, I.L., A.M., R.V., G.G.A. and A.K. Antigen peptide design performed by P.D and S.M. HDX-MS epitope mapping performed by J.A.W and data analysis by J.A.W together with H.P. and S.M. Manuscript lead authors A.M., I.L., S.M. and J.R. All authors have contributed to writing, editing and reviewing the manuscript and results within.

## Funding

## Competing interests

The authors declare the following competing interests: The presented study has been partly funded by Strike Pharma AB, whose long-term value can be influenced by the publication of the paper. S.M., J.R., A.M., I.L., M.L., and P.D. hold private stakes in Strike Pharma AB, whose long-term value can be influenced by the publication of the paper. R.V. is a current employee, and G.G.A. was an employee at Strike Pharma AB when the presented work was carried out. The rest of the authors declare no competing interest.
