## [Transparent Peer Review file · Nature Communications]

A bispecific CD40 agonistic antibody allowing for antibody-peptide conjugate formation to enable cancer-specific peptide delivery, resulting in improved T proliferation and anti-tumor immunity in mice

Corresponding Author: Professor Sara Mangsbo

Version 0:

Reviewer comments:

Reviewer #1

(Remarks to the Author)

The limitations of current antibody-based immunotherapy are evident. Cancer vaccines exhibit immense potential in targeting and eliminating cancer cells by activating the immune system. This manuscript introduces a novel technology called ADAC platform, which enables modular delivery of antigen cargo using antibody drug conjugates. The modularity of this platform is achieved through the interaction between the static pTag sequence and the anti-pTag scFv, which is fused to the parental anti-CD40 architecture. This approach enhances T-cell activation and improves overall survival in mice with tumors. However, certain concerns need to be addressed before publication.

1. On page 15, "SEC-HPLC and accelerated thermal stability study of humanized variants post-incubation at 37°C showed no change in monomeric content compared to untreated samples and the highest T_m for SG variant at 64.3°C (Supp Table 1, Supp Fig. 2)". Please note that the correct table reference for this information should be Supp Table 2.
2. In the figure legend of Supp Fig. 3, the correction should be made in the second line. It should read as "B) SPR analysis of..." instead of "C) ...".
3. No statistical analysis was conducted for Fig. 4D.
4. In the figure legend of Fig. 5C, it is described as "vaccination with the BiA9*2_HF (150 pmol=30 µg) or CpG (20 µg) combined with.....". However, in the figure itself, it is labeled as "BiA9*2_HR+.....". Could you please confirm which variants, BiA9*2_HF or BiA9*2_HR, were actually used in the experiments?
5. The authors administered antibodies at doses of 30 or 100µg for mice treatment. To clarify, does "30 or 100µg" refer to the dosage per kilogram of mice?
6. In Fig 6. D-F, the control group only received the agonistic CD40 antibody (selicrelumab). However, it would be more suitable to include the combination of this antibody with the E7 peptide (44-62aa) as a control to better assess the advantages of ADAC.
7. The utilization of solely the TC-1 tumor model for functional analysis of the ADAC platform is insufficiently persuasive. Is it possible to investigate other tumor models in humanized mouse models with a reconstituted immune system?

Reviewer #2

(Remarks to the Author)

The genesis of the submission is based on application of a modular system to allow cargo delivery. Phage display panning was carried out to identify monoclonal scfv clones against CD40. Humanization of the clone was done to the murine anti p-tag antibody. The antibodies were then engineered to a bispecific format to initiate simultaneous immune activation and

peptide presentation. The study was well designed with a comprehensive analysis.

Comments:

What would be the implication of utilizing the shorter IgG2 hinge on the IgG1 structure in terms of flexibility?

The reference to supp table 1 for thermal stability study of humanized variants should be Suppl Table 2?

Suppl table 1 was not cited in text. Please include.

It would be useful for readers to have a visualization of the bispecific antibody design.

Does the authors have any thoughts as to why the scFv fusion to the CH3 was better than CL? Steric hindrance, accessibility, folding?

Minor:

There are some general typos which can be corrected easily.

The choice of some terms like "Postulated to be thanks to" may not be appropriate.

Reviewer #3

(Remarks to the Author)

Pages / lines are not numbered.

Non-human proteins used in ELISA (Table 2) are of very limited value to determine off-target binding. While this assay may detect severe polyreactivity, IHC on cryosections of various human tissues would be much more conclusive to detect off-target binding relevant to therapeutic applications.

„Two scFv libraries, similar in design to previously described [33], were employed for phage display selection of novel anti-CD40 binders.“ This description is not sufficient to allow a repetition of the experiments described. This style is a general theme throughout the Methods section: e.g. the scFv and IgG expression vectors are not identified.

English language editing may be advised.

Reviewer #4

(Remarks to the Author)

Overview: In this manuscript, the authors describe the construction of a bispecific antibody that targets both CD40 and large T antigen from SV40, and detail some of the biological activity. The intent of the bispecific ab is to build on previous understanding that stimulating CD40 on antigen presenting cells is a potent way of activating them, and that antigen delivered to APC via CD40 can be internalized and cross-presented. The intriguing element to this study is that the second specificity of the molecule is to the pTag antigen, and the authors have the premise that they can deliver bespoke T cell epitopes to APC by making them part of a fusion protein containing a minimal epitope recognized by the T-specific fragment. The study is quite technical in its orientation, demonstrating the development and feasibility of the approach, and some biological efficacy is provided. However, there are several issues with the presentation and data that detract from its message, and ultimately, the manuscript does not advance our understanding of optimal vaccination approaches tremendously.

Critiques:

1. This manuscript is difficult to engage with due to its intense use of specialized nomenclature. For the broad audience of this journal, a more generalized language is recommended. Case in point: "higher post-purification yields and superior end product quality exhibited by BLI and SEC compared to LC fusion variants" with no definition of BLI, SEC or LC. What is A9*1 vs A9*1/2?
2. The premise of the study is poorly presented. It takes a long time to ascertain the intent of the design of the bispecific. No mention of the origin and previous success of targeting tetanus toxoid is presented in the Introduction, and it is very difficult to discern what pTag is actually derived from.
3. Much of the concept of this manuscript was published in the Adv Ther reference #37, limiting the novelty of the approach
4. A tremendous proportion of the manuscript focuses on antibody engineering considerations and product development that likely belong in a more specialized journal.
5. Proportion of CFSEhi vs lo is a very arbitrary assessment of T cell expansion. Number of transferred T cells that have divided and the proportion of undivided is a far better metric.
6. It is unclear why CpG ODN is used as the comparator rather than the exceptionally well characterized FGK45 antibody in Figure 5.
7. Demonstrating ability to generated T cell responses without adoptively transferred TCT transgenic cells is essential
8. Ab+untargeted peptide group is missing from the tumor growth curve in Figure 6B.
9. Evidence of engulfment and direct presentation by cDC would be worthwhile.

Version 1:

Reviewer comments:

Reviewer #1

(Remarks to the Author)

In the revised version, the author has effectively addressed the concerns from the previous review and included new experimental results related to the MC38 model. The study is well-designed and thoroughly analyzed. I believe this paper is suitable for publication in Nature Communications. I have just one minor question:

In Fig. 5C, what do the two bar charts represent individually? Are there any missing labels, such as dLN and ndLN?

Reviewer #2

(Remarks to the Author)

The authors have addressed all the comments. Thank you for an interesting read.

Reviewer #3

(Remarks to the Author)

line 298, typo „array-based screening"

line 712, no data are shown, so the description of results is not sufficient to understand what has been found. Arrays typically show a graded reactivity of multiple proteins (from background to positive control (CD40)). One convincing way to represent this would be to show the signals of background and positive control together with the ten (or more) strongest other reactivities. Only this gives the reader an impression of the quality and quantity of this analysis.

„Kunkel mutagenesis was used to introduce diversity into four of the CDRs (HCDR1-3 and LCDR3).“ If the authors have done any analysis of the resulting diversity (e.g. by sequencing a number of clones), the reader would benefit from an indication of the average number of mutations per CDR, V region or scFv, whatever available.

Reviewer #4

(Remarks to the Author)

I appreciate the clarifications and additional data that the authors have provided. All of my critiques have been adequately addressed by the additional information and the new experiments.

Point to point response reviewers

Reviewer #1 (expert in oncology and cancer drug/vaccine development):

The limitations of current antibody-based immunotherapy are evident. Cancer vaccines exhibit immense potential in targeting and eliminating cancer cells by activating the immune system. This manuscript introduces a novel technology called ADAC platform, which enables modular delivery of antigen cargo using antibody drug conjugates. The modularity of this platform is achieved through the interaction between the static pTag sequence and the anti-pTag scFv, which is fused to the parental anti-CD40 architecture. This approach enhances T-cell activation and improves overall survival in mice with tumors. However, certain concerns need to be addressed before publication.

1. On page 15, "SEC-HPLC and accelerated thermal stability study of humanized variants post-incubation at 37°C showed no change in monomeric content compared to untreated samples and the highest T_m for SG variant at 64.3°C (Supp Table 1, Supp Fig. 2)". Please note that the correct table reference for this information should be Supp Table 2.

Answer: Thank you, this error is now corrected in row 637 in the clean version of the manuscript.

2. In the figure legend of Supp Fig. 3, the correction should be made in the second line. It should read as "B) SPR analysis of..." instead of "C) ...".

Answer: Thank you, this error is now corrected, now in row 1089 in the simple markup version of the manuscript.

3. No statistical analysis was conducted for Fig. 4D.

Answer: Statistical analysis has now been added for Fig. 4D

4. In the figure legend of Fig. 5C, it is described as "vaccination with the BiA9*2_HF (150 pmol=30 µg) or CpG (20 µg) combined with.....". However, in the figure itself, it is labeled as "BiA9*2_HR+.....". Could you please confirm which variants, BiA9*2_HF or BiA9*2_HR, were actually used in the experiments?

Answer: We thank the reviewer for noticing this error and it is now corrected, it was indeed the BiA9*2_HF that was used, row 1016 in the simple markup version of the manuscript.

5. The authors administered antibodies at doses of 30 or 100µg for mice treatment. To clarify, does "30 or 100µg" refer to the dosage per kilogram of mice?

Answer: For clarification the doses indicated in the text and figures refer to the total amount of antibody given per mouse (flat dose), this clarification has been added in the material and methods section row 159 in the simple markup version of the manuscript.

6. In Fig 6. D-F, the control group only received the agonistic CD40 antibody (selicrelumab). However, it would be more suitable to include the combination of this antibody with the E7 peptide (44-62aa) as a control to better assess the advantages of ADAC.

Answer: Very good point, we reasoned that selicrelumab as a clinical comparator is designed to be used as a monoclonal antibody in the clinic, it was thus used as such. Selicrelumab does not include the cargo delivery system using the scFv to carry peptides, thus the appropriate control group is to use the protein that can target CD40 with and without a tag to display the relevance of the cargo delivery in combination with CD40 stimulation with the exact same antibody. We have in both the TC1 model (Figure 6A) along with the new data on the MC38 model (Figure 6H) included the unlinked and linked antigen which in all cases displays superiority with linked antigen.

The manuscript contains the following clarification in the result section, row 762 in clean manuscript file:

“Comparatively, 100 µg of selicrelumab was administered by intravenous infusion as clinically used.”

7. The utilization of solely the TC-1 tumor model for functional analysis of the ADAC platform is insufficiently persuasive. Is it possible to investigate other tumor models in humanized mouse models with a reconstituted immune system ?

Answer: We thank reviewer 1 for this comment, we agree to this and have now included data from the MC38 model, the data is captured in Figure 6 and Supplementary figure 4. As you point out it is of importance to mimic the clinical situation, we therefore chose the MC38 model as there are available documented neoantigens that are of non-viral origin which would further support the concept for clinical translational use.

We felt a risk in using mice with reconstituted immune systems as they can mount strong allogenic responses that may cloud the results.

In the result section row 786 in clean manuscript version we have now stated the following:

“To evaluate if the ADAC technology could lead to improved anti-tumor effects using a true neoantigen-based antibody-peptide conjugate approach, we employed the MC38 model, together with the identified neoantigen Adpgk [40] The selected neoantigen Adpgk has not been shown to promote robust anti-tumor responses in earlier published studies [41] using CpG as adjuvant, interestingly we could confirm that the neoantigen delivered by BiA9*2_HF led to superior anti-tumor control and improved survival compared to a control CpG and peptide treated group (Supp fig. 4F and 4G).”

Reviewer #2 (expert in antibody phage display):

The genesis of the submission is based on application of a modular system to allow cargo delivery. Phage display panning was carried out to identify monoclonal scfv clones against CD40. Humanization of the clone was done to the murine anti p-tag antibody. The antibodies were then engineered to a bispecific format to initiate simultaneous immune activation and peptide presentation. The study was well designed with a comprehensive analysis.

Comments:

1. What would be the implication of utilizing the shorter IgG2 hinge on the IgG1 structure in terms of flexibility?

Answer: We theorized that a more rigid hinge conformation would take form improving the agonistic effect as previously shown by several groups, among them Cragg et al. (DOI: 10.1126/sciimmunol.abm3723). More references to this can be found in the introduction and discussion section of the manuscript with references to nr 12, 37, 41.

2. The reference to supp table 1 for thermal stability study of humanized variants should be Suppl Table 2?

Answer: Thank you, this error is now corrected in row 637 clean manuscript version.

3. Suppl table 1 was not cited in text. Please include.

Answer: , Thank you, supplementary table 1 is now cited in row 526 and 542 in the simple markup version of the manuscript.

4. It would be useful for readers to have a visualization of the bispecific antibody design.

Answer: Thank you for the suggestion and a schematic figure of the bispecific antibody has been added into Figure 4A, where we introduce the bispecific format for the first time.

5. Does the authors have any thoughts as to why the scFv fusion to the CH3 was better than CL? Steric hindrance, accessibility, folding?

Answer: Great question. From a production standpoint the CH3 fusion bispecific exhibited excellent production characteristics compared to LC fusion variants. Generally, a slower translational rate of HC genes in the antibody assembly machinery together with a surplus of LC transcripts have been shown to improve product quality, however the underlying reason is still under debate. With that in mind, we believe the longer mRNA formed by addition of scFv to the c terminal of the

heavy chain requires a longer translation time/mRNA hence “slowing” down the translational rate relative LC translation and skewing the balance towards a more favorable LC:HC ratio in the ER assembly machinery for protein folding respective chain and assembly of the antibody structure. We have added a sentence in the discussion commenting on the better performance of the CH3-fusion (row 856).

Minor:

There are some general typos which can be corrected easily.
The choice of some terms like “Postulated to be thanks to” may not be appropriate.

Answer: Thanks for the comment we have revised the text accordingly, row 858.

Reviewer #3 (expert in antibody phage display):

Pages / lines are not numbered.

1. Non-human proteins used in ELISA (Table 2) are of very limited value to determine off-target binding. While this assay may detect severe polyreactivity, IHC on cryosections of various human tissues would be much more conclusive to detect off-target binding relevant to therapeutic applications.

Answer: Thank you for this remark. We have now added an additional off-target analysis performed with the Retrogenix Cell Microarray Technology, which include 6105 full-length human plasma membrane proteins, secreted and cell surface-tethered human secreted proteins plus a further 400 human heterodimers screened on both fixed and live cells which confirms a high specificity against CD40 and with no off-target binding. This information is added in the Material and Methods and in the result section (row 299 and 711 in the clean manuscript).

2. Two scFv libraries, similar in design to previously described [33], were employed for phage display selection of novel anti-CD40 binders. This description is not sufficient to allow a repetition of the experiments described. This style is a general theme throughout the Methods section: e.g. the scFv and IgG expression vectors are not identified.

English language editing may be advised.

Thank you, we have now updated the method section with the requested information for reproducibility by others. Specifically we have also added a complete sequence list along with information on the germline genes used as library scaffold along with information on that Kunkel mutagenesis was used to introduce diversity:

”Two human synthetic scFv libraries, similar in design and construction to previously described [33], were employed for phage display selection of novel anti-CD40 binders. Briefly, human germline genes IGHV3-23 and IGKV1- 39 were used as

library scaffold and Kunkel mutagenesis was used to introduce diversity into four of the CDRs (HCDR1-3 and LCDR3).“

For clarity we also changed the order of the material and method section to adhere to the order of the illustrated experiments. We also added information on the selection:

“Phagemid DNA from selection rounds 3 and 4 was re-cloned to a vector providing the OmpA signal peptide for secretion of the scFv along with a triple-FLAG tag and a hexahistidine (His6) tag at the C-terminus as described [30]. The constructs were subsequently transformed into TOP10 E. coli. 940 clones were expressed and binding to hCD40 could be confirmed by standard ELISA procedure [30] for 159 of these, of which 59 turned out to be unique”

We hope that this was what was lacking and that we captured the request accurately? If there are more information we should introduce and clarify we are of course happy to add this information.

Reviewer #4 (expert in cancer therapy):

Overview: In this manuscript, the authors describe the construction of a bispecific antibody that targets both CD40 and large T antigen from SV40, and detail some of the biological activity. The intent of the bispecific ab is to build on previous understanding that stimulating CD40 on antigen presenting cells is a potent way of activating them, and that antigen delivered to APC via CD40 can be internalized and cross-presented. The intriguing element to this study is that the second specificity of the molecule is to the pTag antigen, and the authors have the premise that they can deliver bespoke T cell epitopes to APC by making them part of a fusion protein containing a minimal epitope recognized by the T-specific fragment. The study is quite technical in its orientation, demonstrating the development and feasibility of the approach, and some biological efficacy is provided. However, there are several issues with the presentation and data that detract from its message, and ultimately, the manuscript does not advance our understanding of optimal vaccination approaches tremendously.

Critiques:

1. This manuscript is difficult to engage with due to its intense use of specialized nomenclature. For the broad audience of this journal, a more generalized language is recommended. Case in point: “higher post-purification yields and superior end product quality exhibited by BLI and SEC compared to LC fusion variants” with no definition of BLI, SEC or LC. What is A9*1 vs A9*1/2?

Answer: Thank you for this comment. The nomenclature of the antibody and bispecific variants are described in Table 2. BLI and SEC (including meaning of abbreviation) are described in the methods section. Description of “LC” is now added, row 677.

2. The premise of the study is poorly presented. It takes a long time to ascertain the intent of the design of the bispecific. No mention of the origin and previous success of targeting tetanus toxoid is presented in the Introduction, and it is very difficult to discern what pTag is actually derived from.

Answer: Thank you for this, we have updated the manuscript to clarify this. We have explored various tags with the idea that they should be of non-human origin to avoid cross-reactivity, a data set that was too extensive to include. The essence of that work provided us with information on that these tags must retain a certain structure or disordered state regardless of cargo attached, and that the binder and tag must have a high affinity to ensure that the cargo can be carried to the target organ eg lymphoid organs.

Phage display selection gave us binders to tags that varied in their confirmation when bound via biotin to streptavidin selection beads and thus in the end failed to have the properties necessary to provide platform use. Hence, we decided to use a scFv identified via hybridoma generated to a longer identified tetanus toxoid derived B cell epitope that was confirmed to remain in an unstructured state regardless of cargo and attachment, and subsequence trim the tag to a sequence that did not bind endogenous antibodies nor activated T cells to ensure non-immunogenicity of the tag itself.

The following sentence is now included in the manuscript row 183:

“An unstructured universal B cell epitope (18aa) derived from tetanus toxoid was used to generate high affinity antibodies to the same (via immunization and hybridoma generation) and to identify the epitope to which these antibodies bound to. Specifically, the humanized scFv was generated by reformatting the identified murine mAb derived from the hybridoma described in an earlier work from our group[33].”

We are of course also happy to also introduce the work we performed in our quest to screen for high affinity binders to peptide-based tags using phage display, and the failures to identify binders by that method, if requested.

3. Much of the concept of this manuscript was published in the Adv Ther reference #37, limiting the novelty of the approach

Answer: The bispecific design for affinity based antibody conjugate design was described as a proof-of-concept study in the Adv. Therapeutics paper and is based on assessment of two well-known anti-CD40 antibodies. In vitro experiments were performed along with initial in vivo experiments. However, in the in vivo models used in that first paper, the concept was only assessed using transferred human CD40 expressing dendritic cells to wildtype mice. Herein we select out a novel agonistic CD40 antibody and optimize this specific antibody to a clinical candidate with a

selected agonistic antibody with affinity in line the published paper by Yu et al in which they show that agonistic activity is superior in low affinity binders. In addition we strengthen the clinical potential by trimming/optimizing the tag along with humanizing the tag binding scFv, all crucial aspects to enable clinical translational steps.

As the first publication does not have extensive in vivo biological activity in a transgenic hCD40 strain, we also provide further evidence for the affinity conjugate approach to enable its use to treat cancer, in a fully target competent (transgenic human CD40 strain). Apart from tumor data we also provide PK/PD data. In addition we also included cynomolgus data. This data set herein provides a unique stepping stone to introduce a novel antibody drug conjugate format for use in the clinic.

We have now stated this novelty in the introduction row 121 as follows:

“An alternative to traditional ADC-based delivery is to allow for affinity interactions between antibodies and cargo for drug loading (in vitro or in vivo e.g. for theranostics). We earlier identified an alternative coupling strategy to the classical biotin-avidin interaction, which is mainly used as a tool in drug development due to the very immunogenic nature of avidin itself. The identified Adaptive Drug Affinity Conjugate (ADAC) technology was based on a bispecific design carrying an scFv that could bind a synthetic peptide with high affinity, to enable drug cargo delivery to CD40 expressing cells. The goal was to provide a high-affinity interaction with the peptide cargo and thereby allowing for a flexible and rapid cargo loading based on clinical sequencing data and an in-hospital mixing step of patient specific peptides, as a means to provide a dual vaccination and CD40 activating drug entity in one step. Our earlier data provided proof-of-concept data of such technology using a murine-derived scFv and a known B cell epitope (Figure 1A). The reported published data showed extended cargo half-life, targeted cargo delivery and improved T cell activation/expansion in vitro and in vivo (Figure 1A and B) [25]. To reach the clinic the proposed technology requires further work, specifically development of a non-immunogenic tag, a human(ized) binder along with in vivo efficacy data in tumor models. Herein we have identified a novel CD40 agonistic antibody, designed and validated that antibody in the bispecific antibody (BiAb) format applicable to the ADAC platform, enabling antibody-peptide conjugate formation and modular CD40 targeted drug delivery. The study further includes optimization of the antibody format including humanization of the scFv, optimization (trimming) of the tag along with extensive biological efficacy and safety characterization in relevant disease models enabling the step to clinical use.”

4. A tremendous proportion of the manuscript focuses on antibody engineering considerations and product development that likely belong in a more specialized journal.

Answer: We believe the ADAC concept engulfs both the technological and biological aspect of achieving the intended use of the platform. Thus, identifying a novel CD40 agonist, ensuring formatting and activity in the bispecific format is not trivial work, when ensuring that this can be taken to clinical use, thus reinforcing the importance

of the study. The fact that the bispecific format explored herein is also designed to bring a novel antibody format for ADC development, provides uniqueness to the work that has not been published elsewhere other than our first proof of concept study (ref 25). We further would like to reinforce that the additional in vivo data strengthens the biological mode of action concept of the ADAC platform.

5. Proportion of CFSE^{hi} vs lo is a very arbitrary assessment of T cell expansion. Number of transferred T cells that have divided and the proportion of undivided is a far better metric.

Answer: Thank you and we do agree, for clarification, the CFSE^{high} are the undivided and the CFSE^{low} the dividing cells, we have now clarified this in figure 5 and in the figure legend (row 1006).

6. It is unclear why CpG ODN is used as the comparator rather than the exceptionally well characterized FGK45 antibody in Figure 5.

Answer: Thank you for allowing us to clarify this matter. As a general phenomenon, antibodies developed for targeting human CD40 (for clinical use) do not cross-react to murine CD40, which we have also noted for our antibody, hence we make use of a human CD40 transgenic strain that do not express murine CD40. Thus, it is not possible for us to use antibodies targeting murine CD40 as references in the human CD40 transgenic strain. The adjuvant CPG ODN is used as a benchmark comparator since it is well known and commonly used in peptide vaccine strategies in murine models.

7. Demonstrating ability to generate T cell responses without adoptively transferred TCT transgenic cells is essential

Answer: Thank you, we agree. We have included several figures (Figure 3F and Figure 6C) where we demonstrate generating endogenous antigen specific T cell in vivo with the use of the pTag9aa-E744-62 peptide together with the BiA9*2_HF antibody. In these figures the tgCD40 mice were vaccinated three times and the endogenous antigen specific response was evaluated with IFN γ ELISpot or with tetramer staining in whole blood. We have also now performed in vivo experiments with anti-tumor response data using a non-viral neoantigen without adoptive transfer that is added to the revised paper (Fig 6).

8. Ab+untargeted peptide group is missing from the tumor growth curve in Figure 6B.

Answer: Thank you for noticing this error and the requested missing group is now added in figure 6A.

9. Evidence of engulfment and direct presentation by cDC would be worthwhile.

Answer: In our previous study published in *Advanced Technology* we do demonstrate the ability of peptide delivery in vitro into moDCs. This in depth mechanism of the peptide delivery, processing and presentation is part of our current research work and will be published in a separate manuscript.

Point to point response reviewers

Reviewer #1 (expert in oncology and cancer drug/vaccine development):

The limitations of current antibody-based immunotherapy are evident. Cancer vaccines exhibit immense potential in targeting and eliminating cancer cells by activating the immune system. This manuscript introduces a novel technology called ADAC platform, which enables modular delivery of antigen cargo using antibody drug conjugates. The modularity of this platform is achieved through the interaction between the static pTag sequence and the anti-pTag scFv, which is fused to the parental anti-CD40 architecture. This approach enhances T-cell activation and improves overall survival in mice with tumors. However, certain concerns need to be addressed before publication.

1. On page 15, "SEC-HPLC and accelerated thermal stability study of humanized variants post-incubation at 37°C showed no change in monomeric content compared to untreated samples and the highest T_m for SG variant at 64.3°C (Supp Table 1, Supp Fig. 2)". Please note that the correct table reference for this information should be Supp Table 2.

Answer: Thank you, this error is now corrected.

2. In the figure legend of Supp Fig. 3, the correction should be made in the second line. It should read as "B) SPR analysis of..." instead of "C) ...".

Answer: Thank you, this error is now corrected

3. No statistical analysis was conducted for Fig. 4D.

Answer: Statistical analysis has now been added for Fig. 4D

4. In the figure legend of Fig. 5C, it is described as "vaccination with the BiA9*2_HF (150 pmol=30 µg) or CpG (20 µg) combined with.....". However, in the figure itself, it is labeled as "BiA9*2_HR+.....". Could you please confirm which variants, BiA9*2_HF or BiA9*2_HR, were actually used in the experiments?

Answer: We thank the reviewer for noticing this error and it is now corrected, it was indeed the BiA9*2_HF that was used.

5. The authors administered antibodies at doses of 30 or 100µg for mice treatment. To clarify, does "30 or 100µg" refer to the dosage per kilogram of mice?

Answer: For clarification the doses indicated in the text and figures refer to the total amount of antibody given per mouse (flat dose), this clarification has been added in the material and methods section.

6. In Fig 6. D-F, the control group only received the agonistic CD40 antibody (selicrelumab). However, it would be more suitable to include the combination of this antibody with the E7 peptide (44-62aa) as a control to better assess the advantages of ADAC.

Answer: Very good point, we reasoned that selicrelumab as a clinical comparator is designed to be used as a monoclonal antibody in the clinic, it was thus used as such. Selicrelumab does not include the cargo delivery system using the scFv to carry peptides, thus the appropriate control group is to use the protein that can target CD40 with and without a tag to display the relevance of the cargo delivery in combination with CD40 stimulation with the exact same antibody. We have in both the TC1 model (Figure 6A) along with the new data on the MC38 model (Figure 6H) included the unlinked and linked antigen which in all cases displays superiority with linked antigen.

The manuscript contains the following clarification in the result section:

“Comparatively, 100 µg of selicrelumab was administered by intravenous infusion as clinically used.”

7. The utilization of solely the TC-1 tumor model for functional analysis of the ADAC platform is insufficiently persuasive. Is it possible to investigate other tumor models in humanized mouse models with a reconstituted immune system ?

Answer: We thank reviewer 1 for this comment, we agree to this and have now included data from the MC38 model, the data is captured in Figure 6 and Supplementary figure 4. As you point out it is of importance to mimic the clinical situation, we therefore chose the MC38 model as there are available documented neoantigens that are of non-viral origin which would further support the concept for clinical translational use.

We felt a risk in using mice with reconstituted immune systems as they can mount strong allogenic responses that may cloud the results.

In the result section we have now stated the following:

“To evaluate if the ADAC technology could lead to improved anti-tumor effects using a true neoantigen-based antibody-peptide conjugate approach, we employed the MC38 model, together with the identified neoantigen Adpgk [40] The selected neoantigen Adpgk has not been shown to promote robust anti-tumor responses in earlier published studies [41] using CpG as adjuvant, interestingly we could confirm that the neoantigen delivered by BiA9*2_HF led to superior anti-tumor control and improved survival compared to a control CpG and peptide treated group (Supp fig. 4F and 4G).”

Reviewer #2 (expert in antibody phage display):

The genesis of the submission is based on application of a modular system to allow cargo delivery. Phage display panning was carried out to identify monoclonal scfv clones against CD40. Humanization of the clone was done to the murine anti p-tag

antibody. The antibodies were then engineered to a bispecific format to initiate simultaneous immune activation and peptide presentation. The study was well designed with a comprehensive analysis.

Comments:

1. What would be the implication of utilizing the shorter IgG2 hinge on the IgG1 structure in terms of flexibility?

Answer: We theorized that a more rigid hinge conformation would take form improving the agonistic effect as previously shown by several groups, among them Cragg et al. (DOI: 10.1126/sciimmunol.abm3723). More references to this can be found in the introduction section of the manuscript and reference nr 33-38.

2. The reference to supp table 1 for thermal stability study of humanized variants should be Suppl Table 2?

Answer: Thank you, this error is now corrected.

3. Suppl table 1 was not cited in text. Please include.

Answer: Supplementary table 1 is cited in the material and method section but has now also been cited in the text at row 523.

4. It would be useful for readers to have a visualization of the bispecific antibody design.

Answer: Thank you for the suggestion and a schematic figure of the bispecific antibody has been added into Figure 4A, where we introduce the bispecific format for the first time.

5. Does the authors have any thoughts as to why the scFv fusion to the CH3 was better than CL? Steric hindrance, accessibility, folding?

Answer: Great question. From a production standpoint the CH3 fusion bispecific exhibited excellent production characteristics compared to LC fusion variants. Generally, a slower translational rate of HC genes in the antibody assembly machinery together with a surplus of LC transcripts have been shown to improve product quality, however the underlying reason is still under debate. With that in-mind, we believe the longer mRNA formed by addition of scFv to the c terminal of the heavy chain requires a longer translation time/mRNA hence "slowing" down the translational rate relative LC translation and skewing the balance towards a more favorable LC:HC ratio in the ER assembly machinery for protein folding respective chain and assembly of the antibody structure. We have added a sentence in the discussion commenting on the better performance of the CH3-fusion (row 850).

Minor:

There are some general typos which can be corrected easily.
The choice of some terms like “Postulated to be thanks to” may not be appropriate.

Answer: Thanks for the comment we have revised the text accordingly.

Reviewer #3 (expert in antibody phage display):

Pages / lines are not numbered.

1. Non-human proteins used in ELISA (Table 2) are of very limited value to determine off-target binding. While this assay may detect severe polyreactivity, IHC on cryosections of various human tissues would be much more conclusive to detect off-target binding relevant to therapeutic applications.

Answer: Thank you for this remark. We have now added an additional off-target analysis performed with the Retrogenix Cell Microarray Technology, which include 6105 full-length human plasma membrane proteins, secreted and cell surface-tethered human secreted proteins plus a further 400 human heterodimers screened on both fixed and live cells which confirms a high specificity against CD40 and with no off-target binding. This information is added in the Material and Methods and in the result section.

2. Two scFv libraries, similar in design to previously described [33], were employed for phage displayselection of novel anti-CD40 binders.“ This description is not sufficient to allow a repetition of the experiments described. This style is a general theme throughout the Methods section: e.g. the scFv and IgG expression vectors are not identified.

English language editing may be advised.

Answer: We agree with the reviewer and have revised the text accordingly.
In particular more details on phage display selections have been added to support others to repeat the experimental setup.

Reviewer #4 (expert in cancer therapy):

Overview: In this manuscript, the authors describe the construction of a bispecific antibody that targets both CD40 and large T antigen from SV40, and detail some of the biological activity. The intent of the bispecific ab is to build on previous understanding that stimulating CD40 on antigen presenting cells is a potent way of activating them, and that antigen delivered to APC via CD40 can be internalized and cross-presented. The intriguing element to this study is that the second specificity of the molecule is to the pTag antigen, and the authors have the premise that they can deliver bespoke T cell epitopes to APC by making them part of a fusion protein

containing a minimal epitope recognized by the T-specific fragment. The study is quite technical in its orientation, demonstrating the development and feasibility of the approach, and some biological efficacy is provided. However, there are several issues with the presentation and data that detract from its message, and ultimately, the manuscript does not advance our understanding of optimal vaccination approaches tremendously.

Critiques:

1. This manuscript is difficult to engage with due to its intense use of specialized nomenclature. For the broad audience of this journal, a more generalized language is recommended. Case in point: “higher post-purification yields and superior end product quality exhibited by BLI and SEC compared to LC fusion variants” with no definition of BLI, SEC or LC. What is A9*1 vs A9*1/2?

Answer: Thank you for this comment. The nomenclature of the antibody and bispecific variants are described in Table 2. BLI and SEC (including meaning of abbreviation) are described in the methods section. Description of “LC” is now added.

2. The premise of the study is poorly presented. It takes a long time to ascertain the intent of the design of the bispecific. No mention of the origin and previous success of targeting tetanus toxoid is presented in the Introduction, and it is very difficult to discern what pTag is actually derived from.

Answer: Thank you for this, we have updated the manuscript to clarify this. We have explored various tags with the idea that they should be of non-human origin to avoid cross-reactivity, a data set that was too extensive to include. The essence of that work provided us with information on that these tags must retain a certain structure or disordered state regardless of cargo attached, and that the binder and tag must have a high affinity to ensure that the cargo can be carried to the target organ eg lymphoid organs.

Phage display selection gave us binders to tags that varied in their confirmation when bound via biotin to streptavidin selection beads and thus in the end failed to have the properties necessary to provide platform use. Hence, we decided to use a scFv identified via hybridoma generated to a longer identified tetanus toxoid derived B cell epitope that was confirmed to remain in an unstructured state regardless of cargo and attachment, and subsequence trim the tag to a sequence that did not bind endogenous antibodies nor activated T cells to ensure non-immunogenicity of the tag itself.

The following sentence is now included in the manuscript:

“The initial longer tag sequence (18aa) has been identified to be an unstructured universal B cell epitope derived from tetanus toxoid. The humanized scFv was generated by reformatting a murine mAb derived from a hybridoma generated by immunizing mice using a peptide haptentated protein conjugate as previously described [32].”

3. Much of the concept of this manuscript was published in the Adv Ther reference #37, limiting the novelty of the approach

Answer: The bispecific design for affinity based antibody conjugate design was described as a proof-of-concept study in the Adv. Therapeutics paper and is based on assessment of two well-known anti-CD40 antibodies. In vitro experiments were performed along with initial in vivo experiments. However, in the in vivo models used in that first paper, the concept was only assessed using transferred human CD40 expressing dendritic cells to wildtype mice. Herein we select out a novel agonistic CD40 antibody and optimize this specific antibody to a clinical candidate with a selected agonistic antibody with affinity in line the published paper by Yu et al in which they show that agonistic activity is superior in low affinity binders. In addition we strengthen the clinical potential by trimming/optimizing the tag along with humanizing the tag binding scFv, all crucial aspects to enable clinical translational steps.

As the first publication does not have extensive in vivo biological activity in a transgenic hCD40 strain, we also provide further evidence for the affinity conjugate approach to enable its use to treat cancer, in a fully target competent (transgenic human CD40 strain). Apart from tumor data we also provide PK/PD data. In addition we also included cynomolgus data. This data set herein provides a unique stepping stone to introduce a novel antibody drug conjugate format for use in the clinic.

We have now stated this novelty in the introduction as follows:

“Herein we have developed a novel clinical agonistic anti-CD40 candidate and formatted this antibody to a bispecific antibody (BiAb) applicable to the ADAC platform, enabling antibody-peptide conjugate formation and modular CD40 targeted drug delivery. The study further includes optimization of the antibody format including humanization of the scFv, optimization of the tag along with extensive biological efficacy and safety characterization in relevant disease models enabling the step to clinical use. “

4. A tremendous proportion of the manuscript focuses on antibody engineering considerations and product development that likely belong in a more specialized journal.

Answer: We believe the ADAC concept engulfs both the technological and biological aspect of achieving the intended use of the platform. Thus, presenting the development of the bispecific in-context of the application is of importance. We further would like to reinforce that the additional in vivo data strengthens the biological mode of action concept of the ADAC platform.

5. Proportion of CFSEhi vs lo is a very arbitrary assessment of T cell expansion. Number of transferred T cells that have divided and the proportion of undivided is a far better metric.

Answer: Thank you and we do agree, for clarification, the CFSE^{high} are the undivided and the CFSE^{low} the dividing cells, we have now clarified this in figure 5 and in the figure legend.

6. It is unclear why CpG ODN is used as the comparator rather than the exceptionally well characterized FGK45 antibody in Figure 5.

Answer: Thank you for allowing us to clarify this matter. As a general phenomenon, antibodies developed for targeting human CD40 (for clinical use) do not cross-react to murine CD40, which we have also noted for our antibody, hence we make use of a human CD40 transgenic strain that do not express murine CD40. Thus, it is not possible for us to use antibodies targeting murine CD40 as references in the human CD40 transgenic strain. The adjuvant CPG ODN is used as a benchmark comparator since it is well known and commonly used in peptide vaccine strategies in murine models.

7. Demonstrating ability to generate T cell responses without adoptively transferred TCT transgenic cells is essential

Answer: Thank you, we agree. We have included several figures (Figure 3F, Figure 6F and supplementary Figure 4) where we demonstrate generating endogenous antigen specific T cell in vivo with the use of the pTag_{9aa}-E7₄₄₋₆₂ peptide together with the BiA9*2_HF antibody. In these figures the tgCD40 mice were vaccinated three times and the endogenous antigen specific response was evaluated with IFN γ ELISpot or with tetramer staining in whole blood. We have also now performed in vivo experiments with anti-tumor response data using a non-viral neoantigen without adoptive transfer.

8. Ab+untargeted peptide group is missing from the tumor growth curve in Figure 6B.

Answer: Thank you for noticing this error and it is now corrected.

9. Evidence of engulfment and direct presentation by cDC would be worthwhile.

Answer: In our previous study published in Advanced Technology we do demonstrate the ability of peptide delivery in vitro into moDCs. This in depth mechanism of the peptide delivery, processing and presentation is part of our current research work and will be published in a separate manuscript.